# Paired single-cell multi-omics data integration with Mowgli

Geert-Jan Huizing [1,2] ✉, Ina Maria Deutschmann [2], Gabriel Peyré [3] & Laura Cantini [1,2] ✉

The profiling of multiple molecular layers from the same set of cells has recently become possible. There is thus a growing need for multi-view learning methods able to jointly analyze these data. We here present Multi-Omics Wasserstein inteGrative anaLysIs (Mowgli), a novel method for the integration of paired multi-omics data with any type and number of omics. Of note, Mowgli combines integrative Nonnegative Matrix Factorization and Optimal Transport, enhancing at the same time the clustering performance and interpretability of integrative Nonnegative Matrix Factorization. We apply Mowgli to multiple paired single-cell multi-omics data profiled with 10X Multiome, CITE-seq, and TEA-seq. Our in-depth benchmark demonstrates that Mowgli's performance is competitive with the state-of-the-art in cell clustering and superior to the state-of-the-art once considering biological interpretability. Mowgli is implemented as a Python package seamlessly integrated within the scverse ecosystem and it is available at http://github.com/cantinilab/mowgli.

Single-cell sequencing technologies, providing a quantitative and unbiased characterization of cellular heterogeneity, are revolutionizing our understanding of the immune system, development, and complex diseases[1–3]. A new frontier in single-cell sequencing technologies is represented by multi-omics single-cell sequencing, allowing for the simultaneous profiling of multiple molecular readouts (e.g. transcriptome, chromatin accessibility, surface proteins) from the same cell[4–12]. Examples of these cutting-edge sequencing technologies are CITE-seq, simultaneously measuring RNA and surface protein abundance by leveraging oligonucleotide-conjugated antibodies[5], and 10x Genomics Multiome platform, quantifying RNA and chromatin accessibility by microdroplet-based isolation of single nuclei.

Multi-omics single-cell sequencing platforms provide us with complementary molecular readouts from exactly the same set of cells, called in the following paired multi-omics data. The joint analysis of such data offers the exciting opportunity to understand how different molecular facets of a cell collaboratively define the cell's function, morphology, and state[13]. Several multi-view learning methods, jointly analyzing paired multi-omics data by taking into account their shared and complementary information, have thus been recently developed[14–23]. These methods, differently from unpaired integration ones[24,25], take advantage of the known correspondences between cells across modalities. State-of-the-art multi-view learning methods for single-cell multi-omics integration are based on integrative Matrix Factorization[14,19,22], k-nearest neighbors[15], or variational autoencoders[16–18,26,27]. Integrative Matrix Factorization (integrative MF) and variational autoencoders perform dimensionality reduction, jointly embedding the high-dimensional multi-omics cellular profilings into a shared lower-dimensional latent space by leveraging common cells/observations[13,28]. Integrative MF, due to its linear nature, defines a latent space with a natural biological interpretation, but it is too simple to catch complex biological processes[13,28]. On the other hand, nonlinear methods, as variational autoencoders, have shown great potential in clustering cells, but despite recent works on the subject[29,30], they inherently lack biological interpretability. Improving integrative MF methods is thus crucial to striking a balance between interpretability and performance.

We here propose Multi-Omics Wasserstein inteGrative anaLysIs (Mowgli github.com/cantinilab/mowgli), a novel integrative MF

[1]Institut Pasteur, Université Paris Cité, CNRS UMR 3738, Machine Learning for Integrative Genomics Group, F-75015 Paris, France. [2]Institut de Biologie de l'Ecole Normale Supérieure, CNRS, INSERM, Ecole Normale Supérieure, Université PSL, 75005 Paris, France. [3]CNRS and DMA de l'Ecole Normale Supérieure, CNRS, Ecole Normale Supérieure, Université PSL, 75005 Paris, France. ✉e-mail: geert-jan.huizing@pasteur.fr; laura.cantini@pasteur.fr

method for single-cell multi-omics data combining integrative Non-negative Matrix Factorization[31] (integrative NMF) with Optimal Transport[32] (OT). On one hand, Mowgli employs integrative NMF, popular in computational biology due to its intuitive representation by parts and further enhances its interpretability[31]. On the other hand, Mowgli enhances the clustering performances of integrative MF by taking advantage of OT, which we have previously shown to better capture similarities between single-cell omics profiles[33]. We extensively benchmarked Mowgli with respect to the state-of-the-art in the integration of several paired multi-omics data profiled with CITE-seq[5], 10X Genomics Multiome, and TEA-seq[7] platforms. Of note, while we focused on the integration of the currently available omics data, Mowgli can deal with paired multi-omics datasets with any type and number of omics, without any statistical assumption on the data. The performed in-depth comparison showed that Mowgli's embedding and clustering quality outperform the state-of-the-art in controlled settings derived from real multi-omics data and are competitive with the state-of-the-art in more complex real multi-omics data. Of note, the latter are affected by the lack of an absolute ground truth annotation on most real datasets. Finally, Mowgli was shown to improve the state-of-the-art in terms of biological interpretability through an in-depth biological analysis of TEA-seq data.

## Results

### Mowgli: a new tool for paired single-cell multi-omics data integration

We developed Multi-Omics Wasserstein inteGrative anaLysIs (Mowgli), a new tool for paired single-cell multi-omics data integration (github. com/cantinilab/mowgli).

Mowgli is based on integrative Matrix Factorization (integrative MF). Starting from $d$ omics matrices $\mathbf{A}^{(p)} \in \mathbb{R}^{m_p \times n}$ with $p \in [1 \ldots d]$, sharing the same columns (the cells) but having different features (e.g. genes, peaks), Mowgli jointly decomposes them into the product of omic-specific dictionaries $\mathbf{H}^{(p)} \in \mathbb{R}^{m_p \times k}$ and a shared embedding $\mathbf{W} \in \mathbb{R}^{k \times n}$ with $k \ll m_p$ and $k \ll n$ (Fig. 1A). As a standard nomenclature, in the following we will call k the number of latent dimensions, the columns of $\mathbf{H}^{(p)}$ loadings and the rows of $\mathbf{W}$ factors[28,34].

In line with state-of-the-art MF methods for multi-omics integration[35], the cell embedding $\mathbf{W}$ can be used to visualize and cluster the cells (Fig. 1B)[36–39]. The dictionaries $\mathbf{H}^{(p)}$ instead enable biological interpretation via gene set enrichment analysis[40], motif enrichment analysis[41], or by identifying markers among the top weights (Fig. 1C).

The main innovation of Mowgli is to perform integrative MF by combining integrative Non-Negative Matrix Factorization (integrative NMF) with Optimal Transport (OT). It thus solves the optimization

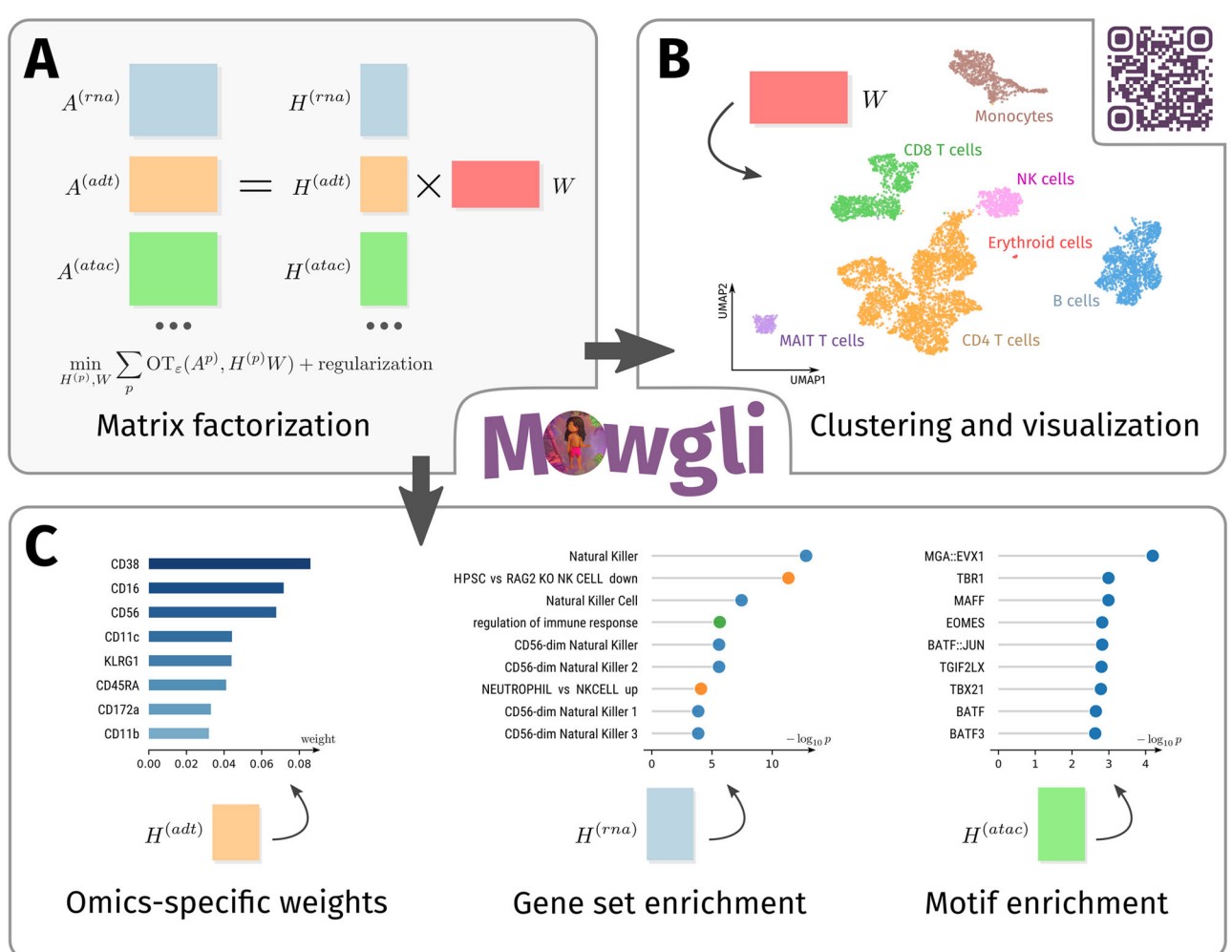

**Fig. 1 | Overview of Mowgli. A** Schematic visualization of Mowgli, an NMF-based model with an Optimal Transport loss; **B** the matrix $W$ of Mowgli can be used for cell clustering and visualization; **C** The dictionaries $\mathbf{H}^{(p)}$ of Mowgli contain omics-specific weights for each latent dimension, which can be used for the biological characterization of the latent dimensions through gene set enrichment or motif enrichment analysis.

problem:

$$\min_{\mathbf{H}^{(p)},\mathbf{W}} \sum_p OT_\varepsilon\left(\mathbf{H}^{(p)}\mathbf{W},\mathbf{A}^{(p)}\right) - \rho_p E\left(\mathbf{H}^{(p)}\right) - \mu E(\mathbf{W}) \text{ with omics } p = 1,\ldots,m \quad (1)$$

In computational biology, integrative NMF is usually applied with an Euclidean reconstruction term between $\mathbf{H}^{(p)}\mathbf{W}$ and $\mathbf{A}^{(p)}$[19,22,24]. We here introduce instead a reconstruction term based on entropy-regularized Optimal Transport (OT) (see Eq. 1 and Methods), which unlike Euclidean or Kullback-Leibler losses leverages a notion of similarity between features. This choice is justified by the improved performance that we have previously observed once using OT to compare single-cell omics profiles[33]. Of note, outside of biology, OT has been already used in the reconstruction loss of NMF for the factorization of single matrices[42–44] and single tensors[45].

In addition, as in Rolet et al.[42], we add to the optimization problem (Eq. 1) two entropic regularization terms $\rho_p E(\mathbf{H}^{(p)})$ and $\mu E(\mathbf{W})$ (see Methods). These terms ensure that the loadings and embeddings are positive distributions and they control their sparsity (see Methods), a crucial feature to further enhance the known NMF's representation by parts and sparsity properties[31]. $\rho_p$ and $\mu$ are the coldness parameters of softmax functions (see Methods) and thus offer a natural way to adjust sparsity. For instance, as $\mu$ approaches 0, cells will be assigned to only one factor. As instead $\mu$ increases, cells will be a combination of several factors. For all details on the mathematical formulation of Mowgli see Methods.

Of note, Mowgli is implemented as an open-source Python package seamlessly integrated into the classical Python single-cell analysis pipeline (github.com/cantinilab/mowgli). Users can thus take advantage of scverse tools like Scanpy and Muon for preprocessing and downstream analysis[46,47]. In addition, Mowgli provides a user-friendly visualization of top genes and enriched gene sets, thus helping biological interpretability.

In the following, we extensively benchmark Mowgli against the state-of-the-art: Seurat v4[15], Cobolt[26], Multigrate[27], and MOFA+[14]. Although several methods exist[14–23], we here focused on the leading methods for paired data integration that could be applied to the multiple combinations of single-cell omics data here considered. In addition, an integrative NMF baseline is also considered (see Methods), to further compare Mowgli with the standard integrative NMF.

## Mowgli's cell embedding and clustering outperform the state-of-the-art in controlled settings derived from cell lines data

We first focused on evaluating Mowgli's embedding and clustering performance in controlled settings derived from cell lines data. To represent a panel of realistic scenarios with different distributions of cells across three groups, we applied different transformations to a simple dataset composed of three cancer cell lines profiled with scCATseq (see Fig. 2A). The scCATseq dataset provides a joint profiling of scRNA and scATAC from HCT116, HeLa-S3, and K562 cell lines[8]. Unlike simulated data, this solution allows us to avoid making assumptions about the distribution of the data. Indeed, generating simulated data following a Gaussian distribution, for instance, would favor methods that approximate single-cell data with this same distribution.

The four scenarios in our panel represent distinct realistic challenges of multi-omics integration: (i) *Mixed in RNA* contains two cell populations that are mixed in scRNA, but well separated in scATAC; (ii) *Mixed in both* contains two cell populations mixed in scRNA and well separated in scATAC and two cell populations mixed in scATAC

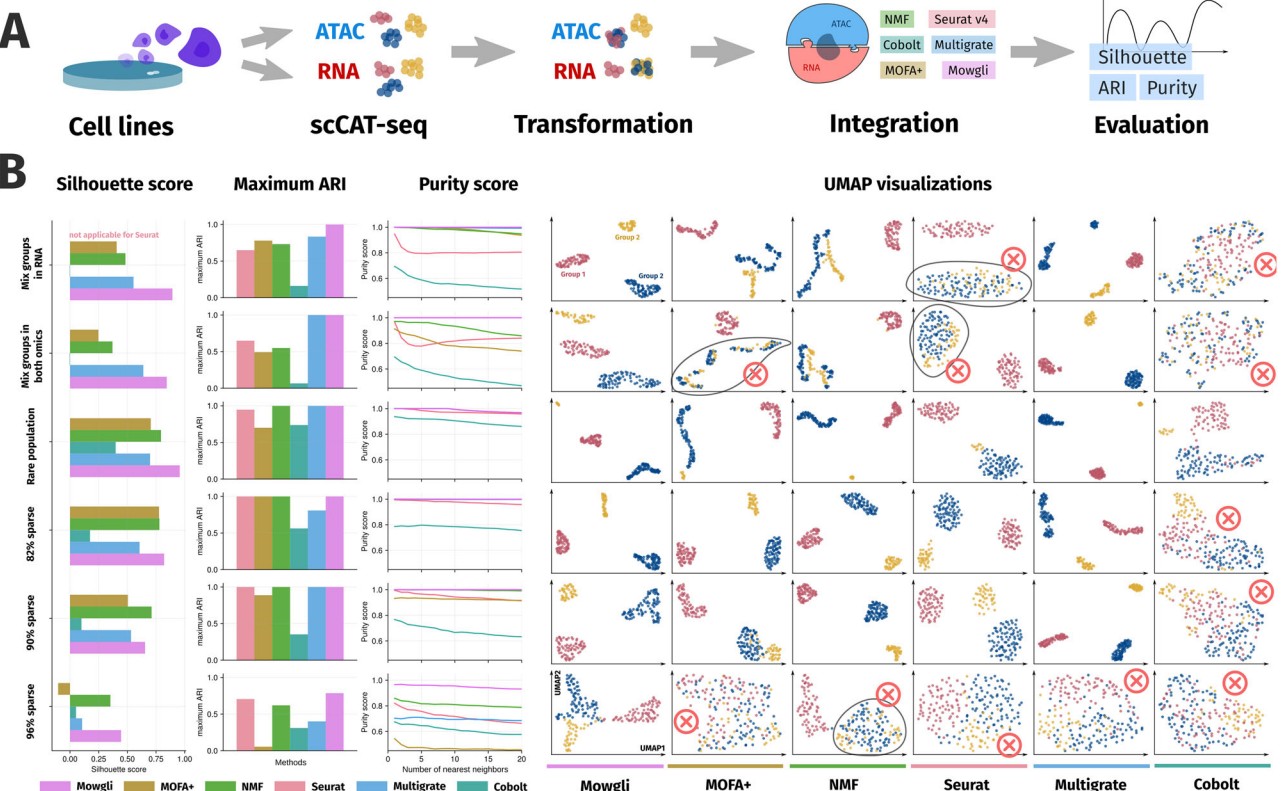

**Fig. 2 | Cell embedding and clustering benchmark in controlled settings derived from cell lines data. A** Schematic representation of the benchmarking process; **B** The first three columns of this panel are devoted to silhouette scores, Adjusted Rand Indices (ARIs), and purity scores for the different methods on six controlled settings derived from cell lines data. The following six columns provide UMAP visualizations for the six benchmarked methods (Mowgli, MOFA+, NMF, Seurat v4, Multigrate, Cobolt) on six controlled settings derived from cell lines data. Different colors in these UMAP plots correspond to the three groups of cells imposed in the dataset.

and well separated in scRNA; (iii) *Rare population* presents a population with much fewer cells than the others and (iv) 82% sparse, 90% sparse, and 96% sparse contain data with increasing percentages of dropouts (82–96%). Scenarios (i) and (ii) test the ability of methods to take into account the complementarity of different omic data. Scenario (iii) tests the ability of the methods to recover rare populations. Finally, scenario (iv) tests the robustness of the methods to dropout noise, while staying in a realistic range of dropouts for single-cell data. For details on the generation of these datasets see Methods.

We benchmarked Mowgli, Seurat v4, MOFA+, integrative NMF, Cobolt, and Multigrate based on natural metrics for embedding and clustering performance: silhouette score, Adjusted Rand Index (ARI), and purity score (see Methods). The silhouette score cannot be computed for Seurat because it requires an embedding of the cells not provided by this method. In addition, we computed UMAP visualizations for the different methods and datasets[37].

As shown in Fig. 2B, overall, Mowgli provides superior performance over the current state-of-the-art according to all metrics. Indeed, in all datasets except 90% *sparse*, Mowgli has a performance greater or equal to that of other methods. In the *90% sparse* dataset, integrative NMF has a better silhouette score than Mowgli but the same ARI and purity score. Cobolt performs poorly in *Mixed in RNA* and *Mixed in both* according to all three metrics.

These performances are confirmed by looking qualitatively at the UMAP plots in Fig. 2B. In *Mixed in RNA* and *Mixed in both* Seurat v4 and Cobolt confuse populations when individual omics are not sufficient to identify the three groups. Regarding dropouts, one of the most challenging features of single-cell data[48], Mowgli shows the highest resilience with respect to the state-of-the-art. Indeed, while a sparsity of

96% is still coherent with realistic data[49], MOFA+, Seurat v4, Cobolt, and Multigrate confuse the three populations in the 96% *sparse* dataset. On the opposite, Mowgli correctly separates the three groups of cells in 96% sparse.

All methods except Seurat are parametrized by a number of latent dimensions. For each method, we choose the overall best-performing number of latent dimensions (see Methods). Tuning the number of latent dimensions for each metric and dataset does not change our conclusions (see Supplementary Fig. 1).

## Mowgli's cell embeddings and clusterings are competitive with the state-of-the-art in complex and heterogeneous datasets

We then benchmarked Mowgli, Seurat v4, MOFA+, integrative NMF, Cobolt, and Multigrate based on their embedding and clustering performance on five paired single-cell multi-omics datasets (see Fig. 3A). Of note, these data have been already largely used to benchmark single-cell multi-omics integrative methods[15,50,51]. The chosen datasets span different sequencing technologies, modalities, tissues, and sizes: (i) Liu is a scCAT-seq cell lines dataset by Liu et al.[8] (ii) *PBMC 10X* is a 10X Multiome human PBMC dataset from 10X Genomics (iii) *OP Multiome* is a 10X Multiome human bone marrow dataset from Open Problems[52] (iv) *OP CITE* is a CITE-seq human bone marrow dataset from Open Problems[52] (v) *BM CITE* is a CITE-seq human bone marrow dataset from Stuart et al.[25]. *BM CITE* is the larger dataset here considered, with 29,803 cells. Supplementary Table 1 lists the modalities, numbers of cells, and numbers of cell types for each dataset. For details on data preprocessing, see Methods.

We benchmarked Mowgli, Seurat v4, MOFA+, integrative NMF, Cobolt, and Multigrate based on the same natural metrics used in the

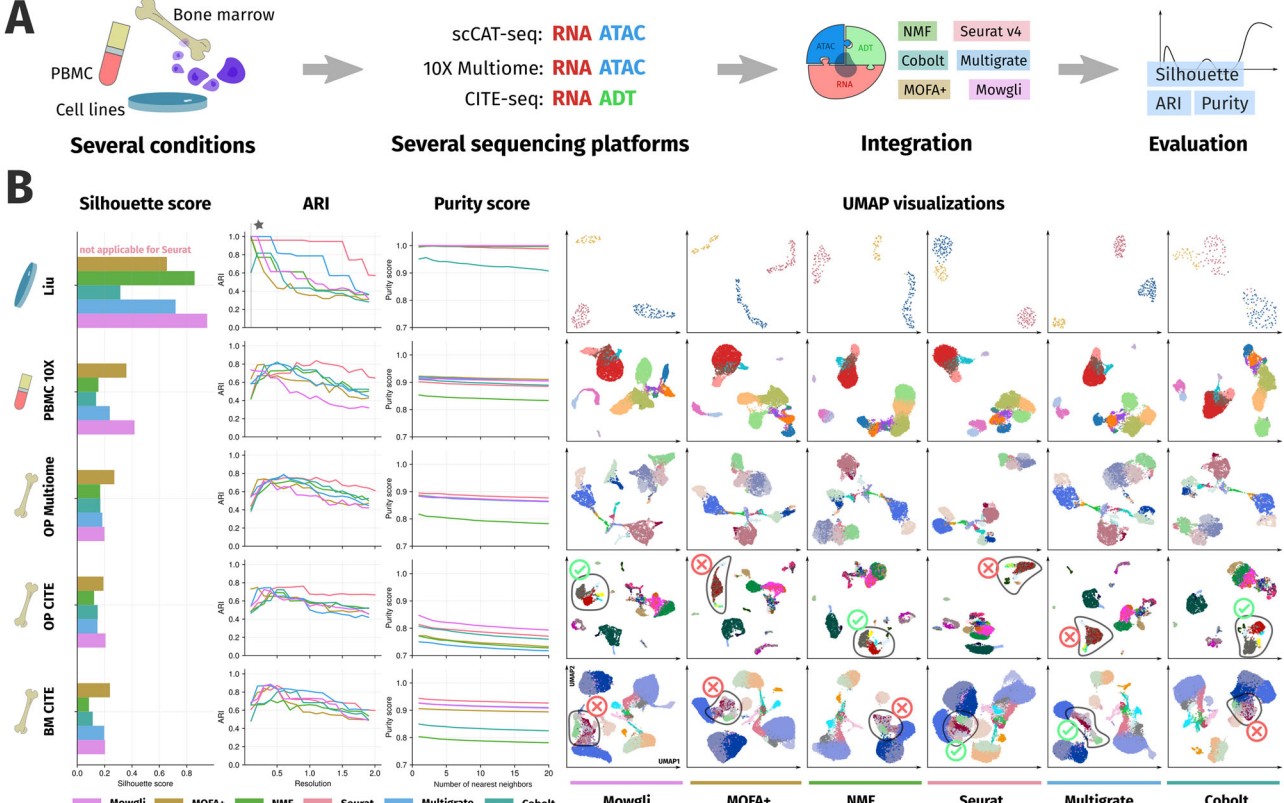

**Fig. 3 | Cell embedding and clustering benchmark in complex and heterogeneous datasets.** **A** Schematic representation of the benchmarking process; **B** The first three columns of this panel are devoted to silhouette scores, Adjusted Rand Indices (ARIs), and purity scores for the different methods on five complex paired single-cell multi-omics data already largely used to benchmark integrative

methods. The following six columns provide UMAP visualizations for the six benchmarked methods (Mowgli, MOFA+, NMF, Seurat v4, Multigrate, Cobolt) on the same data. Different colors in these UMAP plots correspond to the different ground truth cell type annotations provided with the data.

previous section. Since these metrics require a ground truth annotation, we used the cell-type annotations available from the original publications of these data. In Liu, the ground truth annotations are based on the cell line of origin and are thus well-defined. On the contrary, the annotations of the other datasets were computationally derived, thus affecting this benchmark for all methods. Supplementary Note 1 illustrates this on specific subpopulations of CD8 T-cells in the *BM CITE* and *OP CITE* datasets.

As displayed in Fig. 3B, Mowgli can handle large single-cell datasets and deliver embedding and clustering performances competitive with the state-of-the-art, especially considering the lack of absolute ground truth annotations on most datasets here employed.

In particular, according to the silhouette score, Mowgli outperforms other methods in *Liu*, *PBMC 10X*, and *OP CITE*. MOFA+ performs best in the other two datasets. In terms of ARI across resolutions, Seurat v4 performs best in *PBMC 10X*, *OP Multiome*, and *OP CITE*. Mowgli and Seurat v4 perform comparably in the *BM CITE* dataset. In the *Liu* dataset, only MOFA+, Multigrate, and Mowgli reach a maximum ARI of 1. Of note, in *Liu*, ARIs should be compared only at low resolution, as higher resolutions lead to overclustering. In terms of purity score, Mowgli outperforms other methods in the *OP CITE* dataset, and it is comparable to MOFA+ and Multigrate in the *PBMC 10X* dataset. Finally, the purity scores of all methods are comparable in the *Liu* dataset, except for Cobolt which is less performant.

The UMAP plots in Fig. 3B give a qualitative intuition of the described performance. In *OP CITE*, only integrative NMF, Cobolt, and Mowgli correctly separate subpopulations of B cells (Fig. 3B circled). In *BM CITE*, MAIT T-cells and subpopulations of CD8+T-cells (Fig. 3B circled) are more neatly separated in Seurat v4 than in other methods. However, as explained in Supplementary Note 1, the annotation pipeline of *BM CITE* might favor Seurat v4.

As in the previous section, we chose the overall best-performing number of latent dimensions for each method (see Methods). Tuning the number of latent dimensions for each metric and dataset does not favor one method over the others, which supports Mowgli's competitiveness with the other methods (see Supplementary Fig. 1).

## Mowgli improves the biological interpretability of the state-of-the-art by providing cell-type specific factors in TEA-seq data

We benchmarked Mowgli with respect to MOFA+ and integrative NMF based on its biological interpretability (see Fig. 4A). Indeed, MOFA+ is the leading single-cell multi-omics integration tool providing user-friendly biological interpretability of its latent dimensions[14]. At the same time, integrative NMF can be considered as a baseline with respect to Mowgli.

For this benchmark, we considered a TEA-seq dataset of human PBMCs, corresponding to the paired profiling of scRNA-seq, scATAC-seq, and surface proteins[7]. This dataset allows us to test the methods on more than two omic datasets, thus taking into account more complementary layers of molecular regulation.

First, MOFA+, integrative NMF, and Mowgli were independently applied for the integration of the three omics constituting the TEA-seq data. As the dataset was not provided with an annotation of the cells, we separately clustered the embeddings obtained from Mowgli, integrative NMF, and MOFA+ and annotated them based on gene and protein markers (see Supplementary Fig. 2, see Fig. 4B). We identified in this way coarse immune cell types: CD4 T-cells, CD8 T-cells, B cells, Natural Killer (NK) cells, MAIT T-cells, Monocytes, and Erythroid cells. Of note, the cell type annotations obtained with Mowgli, integrative NMF, and MOFA+ agree at 94% and match an independent RNA-based annotation obtained through Azimuth (see Supplementary Fig. 3). All three methods are thus able to recover the expected cell types through clustering of their embeddings.

To then test the biological interpretability of Mowgli, integrative NMF, and MOFA+, we evaluated the specificity of the associations between their factors and the identified immune cell types. The underlying assumption we are making here is that an interpretable method should provide factors that are not broadly active in all the cells, but selectively associated with a cell type. Indeed, characterizing a cell type that results from a combination of many factors is a daunting task. On the contrary, having cell type-specific factors makes the biological characterization of the associated cell type straightforward. To evaluate such specificity, for each cell type, we plotted how Mowgli, integrative NMF, and MOFA+'s factors are distributed according to their mean weight within the cell type and their mean

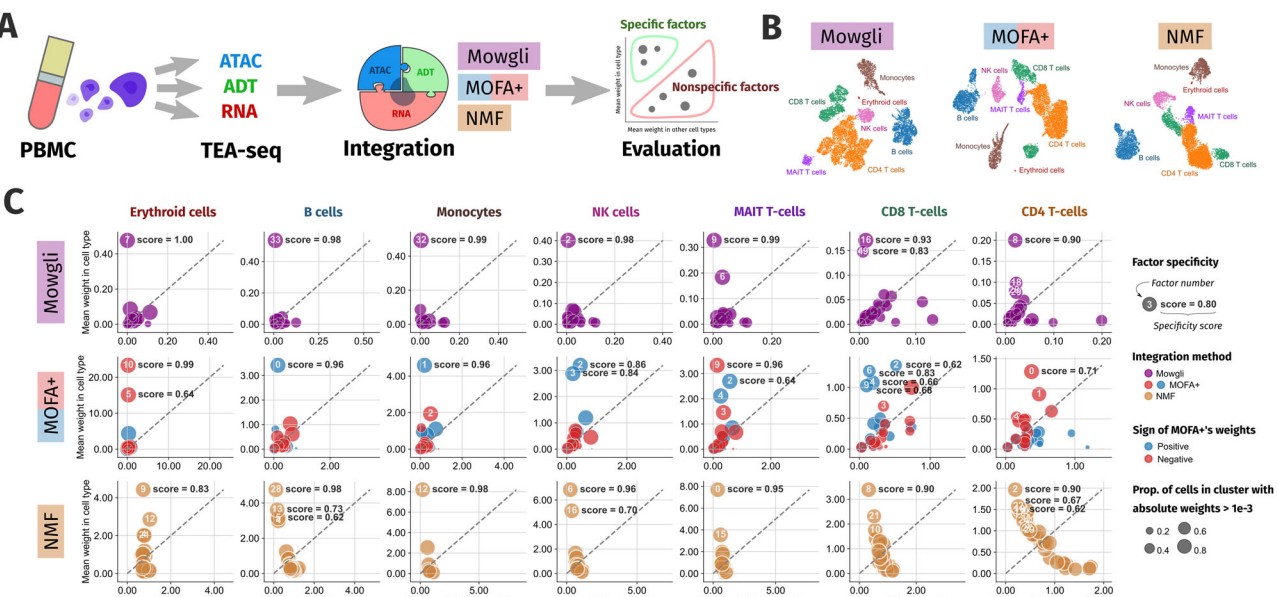

**Fig. 4 | Evaluation of biological interpretability in TEA-seq data. A** Schematic representation of the evaluation process on biological interpretability; **B** UMAP visualization of Mowgli's, MOFA + 's, and integrative NMF's embeddings. The colors correspond to a marker-based cell-type annotation of the cells; **C** average weights within and outside of a cell type are plotted for each factor of Mowgli (violet), MOFA+ (red for the negative part and blue for the positive one), and integrative NMF (orange). For each cell type, the best specificity scores are reported in bold.

weight outside the cell type (Fig. 4C). Factors specific to a cell type should have a high average weight within the cell type and a low average weight outside the cell type, thus falling in the upper left corner of the plots. As MOFA+'s factors are not constrained to be positive and their positive and negative parts could be associated with different biological information, we split each factor into two parts, as done in MOFA+'s interpretation tools[14]. In addition, we quantified the performance of each factor with a specificity score, also reported in bold in Fig. 4C, and defined in the Methods section.

As shown in Fig. 4C, while MOFA+ and integrative NMF tend to associate multiple factors to the same cell type, Mowgli frequently defines clear one-to-one associations between factors and cell types. In addition, the specificity score of such factors is higher in Mowgli than in MOFA+ and integrative NMF. This is particularly striking in NK cells, CD8 T-cells, and CD4 T-cells, where both MOFA+ and integrative NMF seem to aggregate information from many factors whereas Mowgli is more selective. Of note, as shown in Supplementary Fig. 4, the multiple factors associated by MOFA+ with the same cell type do not necessarily correspond to subpopulations of the same cell type.

As shown in Supplementary Fig. 5, our conclusions regarding the specificity of Mowgli's factors are robust to the choice of hyperparameters.

Overall, the comparison with integrative NMF validates that the use of OT as a loss function and of sparsity-inducing regularization introduced in Mowgli has a concrete effect on biological interpretability.

## Mowgli identifies relevant subpopulations of immune cells in TEA-seq data

We finally focused on the biological relevance of the factors identified by Mowgli on the human PBMC TEA-seq data, described in the previous section. Indeed, while in the previous section we only considered coarse immune cell types (e.g. B cells, CD4 T-cells, CD8 T-cells), Mowgli could identify multiple factors able to subset such cell types into relevant subpopulations (see Fig. 5A, B; Supplementary Fig. 6; Supplementary Note 2). For example, Mowgli identifies factors splitting the B cell cluster into two subpopulations: memory and naive B cells. In the same way, Mowgli detects factors associated with CD8 T-cells subpopulations (naive, central memory, and effector memory), monocytes subclusters (classical and non-classical), dendritic cells subpopulations (plasmacytoid and conventional) and Natural Killer (NK) subclusters (CD56$^{dim}$ and CD56$^{bright}$). The association of the factors with specific immune subpopulations is here made based on top-ranked genes and proteins in effector memory CD8 T cells, naive B cells, memory B cells, and CD56$^{dim}$ NK cells. For all other populations, the association with factors is instead based on the correlation of the factors' weights with that of known protein markers (see Supplementary Note 2). Figure 5B displays side-by-side the UMAP plots showing the similarity between the distribution of the factors' weights and the activity of the protein markers of their associated immune subpopulations. The UMAP visualizations of all factors and marker proteins are available in Supplementary Fig. 6 and Supplementary Fig. 7.

These same results could not be obtained with MOFA+, due to its lower biological interpretability observed in the previous section. In MOFA+, factors having similar patterns to those observed in Mowgli could be obtained for effector memory CD8 T-cells, memory B cells, non-classical monocytes, and CD56dim NK cells (see Supplementary Fig. 4 and Supplementary Fig. 8). On the contrary, MOFA+'s factors most closely associated with the other immune subpopulations identified by Mowgli have less clear patterns (see Supplementary Note 2). As a consequence, interpreting with MOFA+ the pathways associated with CD56$^{bright}$ NK cells, for example, would require complexly combining the pathway enrichments obtained from different factors. On the contrary, the same analysis in Mowgli can be easily realized by looking at the pathways enriched in the loadings of its 13th factor.

Finally, we looked at the biological information that Mowgli could provide regarding the identified immune subpopulations. For this part, we focused on the factors associated with four immune cell subpopulations: effector memory CD8 T-cells (factor 49), naive B cells (factor 33), memory B cells (factor 44), and CD56$^{dim}$ NK cells (factor 2). For each of these four factors, we considered their associated loadings in $\mathbf{H}^{(rna)}$, $\mathbf{H}^{(adt)}$ and $\mathbf{H}^{(atac)}$ and analyzed the top genes in $\mathbf{H}^{(rna)}$, top proteins in $\mathbf{H}^{(adt)}$, the gene sets enriched in $\mathbf{H}^{(rna)}$ and the motifs enriched in $\mathbf{H}^{(atac)}$ to verify the biological information that could be extracted from the output of Mowgli (see Methods). Figure 5C displays the results obtained from this analysis.

For effector memory CD8 T-cells (CD8 TEM cells), corresponding to factor 49, Mowgli could extract two top genes (*CRTAM* and *KLRK1*), known to be essential for CD8+T-cell-mediated cytotoxicity[53,54], two top proteins (CD45RO, TCR-a/b) that are a known memory T cell marker and a T cell receptor, respectively[55,56]. More interestingly, also several Transcription Factors (TFs) candidate regulators of this subpopulation are identified, among them EOMES and TBX21 (aka T-bet), known to be important for CD8 TEM development[57]. In addition, five of the top candidate TF regulators (TBR1, TBX21, TBX4, TBX5 and MGA) target three of the top genes of the same factor (*CCL5*, *CRTAM*, and *IL21R*), thus suggesting a regulatory program possibly important for CD8 TEM cells.

In naive B cells (factor 33), Mowgli identifies as top genes *FCER2* (aka *CD23*), a low-affinity receptor for immunoglobulin E (IgE) with an essential role in the differentiation of B-cells[58] and *MARCH1*, which downregulates the surface expression of major histocompatibility complex (MHC) class II molecules[59]. In the top proteins, we can single out CD19, CD21, and HLA-DR, well-known markers of B cells[60]. In addition, the relative weights of IgD and IgM in factor 33 are coherent with the repartition already described for naive B-cells[60]. Finally, among the top TF candidate regulators of factor 33, Early B-cell Factors (EBF3 and EBF1)[61] and NF-kB proteins (REL and RELA) stand out as regulators of the top genes of the same factor. Of note, these TFs play an essential role in B-cell development, maintenance, and function[62].

For memory B cells (factor 44), Mowgli extracts as top genes: *IGHA1* and *IGKC*, part of immunoglobulin complexes[63] and *JAM3*, belonging to the Immunoglobulin superfamily and already studied in the context of B cell homing and development[64–66]. The top proteins include the well-known B cell markers CD19, CD21, and HLA-DR[60]. In addition, as observed before for naive B cells, the relative weights of IgD and IgM in factor 44 are coherent with the repartition already described for memory B-cells[60]. In the top TFs emerging from our motif analysis and targeting the top genes we finally find RELA, TCF4, and MAX::MYC, known to be involved in the transcriptional regulation of memory B cell differentiation[67].

Finally, in CD56$^{dim}$ NK cells (factor 2), Mowgli detects at top genes: *NCAM1* (aka *CD56*), the go-to marker for NK cells[68]; *KLRF1* and *KLRD1*, genes of the KLR family of receptors controlling NK cell activity[69]; *GZMB*, involved in NK-cell mediated cytotoxicity[70]; *SLAMF7*, mediating NK cell activation[71]. Top proteins include CD56, the canonical marker of NK cells[68], but its weight is lower than that of CD16, which is coherent with the expression profile of CD16+CD56$^{dim}$ NK cells[68]. Regarding TF candidate regulators, we detect EOMES and TBX21 (aka T-bet), which are critical to NK-cell differentiation[72], Maf-F, having a key role in the regulation of NK cell effector functions by IL-27, and JUNB::FOSB, early activator protein (AP)−1 TFs that regulate NK-mediated cytotoxicity[73,74]. Finally, a strong regulatory program seems to emerge here with four of the top candidate TF regulators for factor 2 (MGA::EVX1, EOMES, TBX21, and JDP2) targeting four of the top genes of the same factor (*C1orf21*, *IL18RAP*, *PTGDR*, and *SLAMF7*).

## Discussion

Multiple technologies allowing the multi-omics profiling of the same set of cells are currently available. We thus need integration methods

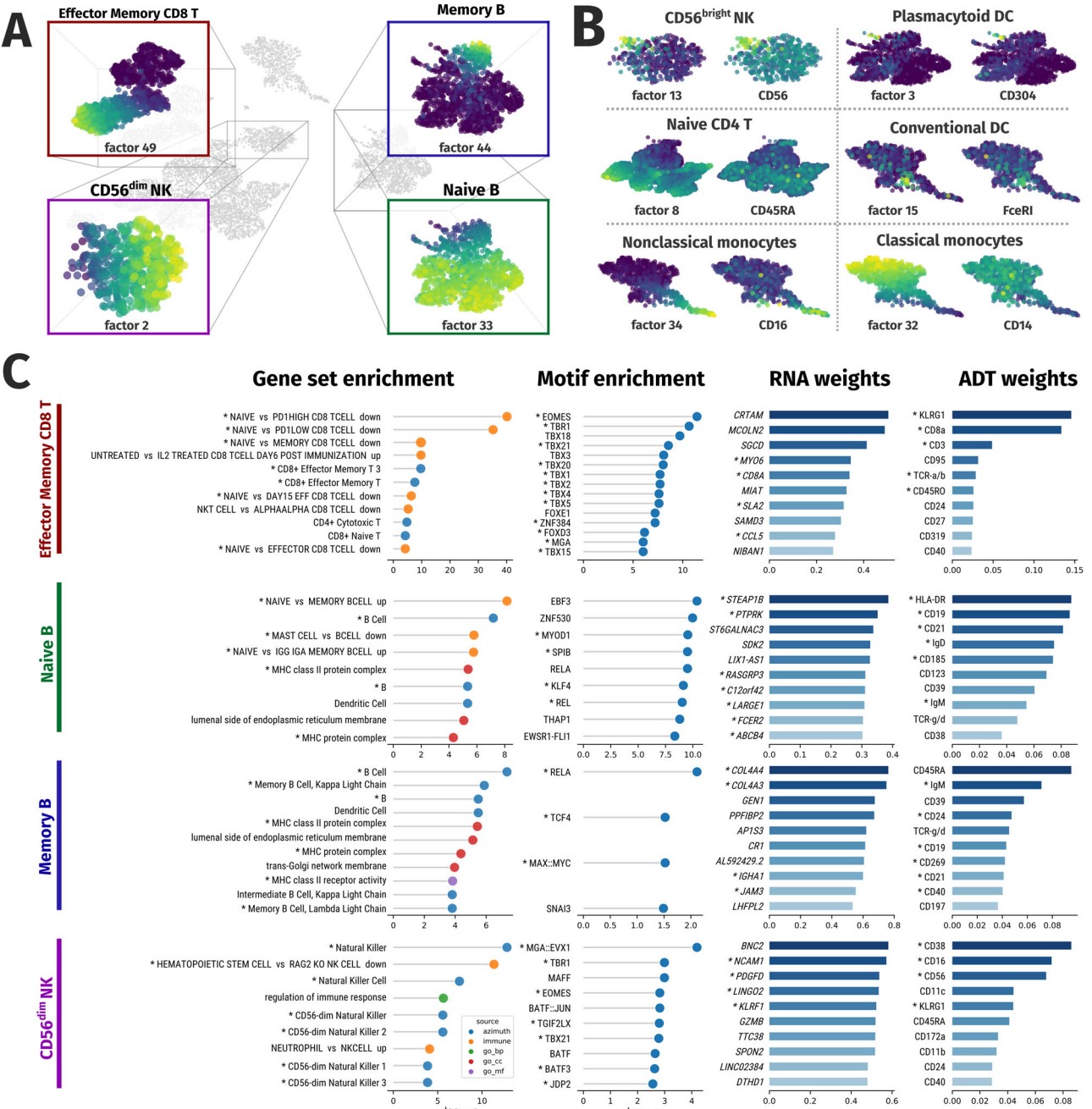

**Fig. 5 | Characterization of the immune cell subpopulations identified by Mowgli in TEA-seq data. A** UMAP visualization of Mowgli's embedding with focus on four specific immune subpopulations (Effector Memory CD8 T-cells, memory B cells, CD56$^{dim}$ NK cells, naive B cells) for which the UMAP is colored based on factor weights; **B** UMAP visualization of Mowgli's embedding colored by factor weight and protein marker weight for other factors corresponding to specific subpopulations of cells; **C** Top genes, proteins, gene sets and Transcription Factors (TFs) for the 4 factors visualized in panel a. Stars denote gene sets and markers pertinent for the immune subpopulation associated with the factor and TFs targeting the top genes.

able to jointly learn from multiple omics data profiled on the same cells.

In this article, we introduced Multi-Omics Wasserstein integrative anaLysIs (Mowgli), an integrative method for paired multiomics data that enables rich biological interpretation for any type and number of omics. We then in-depth benchmark Mowgli's cell embedding and clustering performance with respect to the state-of-the-art in controlled settings derived from scCAT-seq profiling of cancer cell lines. Mowgli outperforms in this benchmark the state-of-the-art showing its high potential even in challenging conditions. We then considered more complex and heterogeneous data profiled

with CITE-seq and 10x Genomics Multiome technologies. On these data, Mowgli performed comparably with the state-of-the-art, with no method clearly outperforming others. Finally, regarding the biological interpretability, once tested on TEA-seq data, corresponding to paired scRNA, scATAC, and surface protein profiling, Mowgli produces biologically meaningful representations superior to those of the state-of-the-art.

A major limitation affecting this benchmark and all others focused on paired multi-omics integration corresponds to the lack of a high-quality biological annotation of the cells. While in some cases Fluorescence-activated cell sorting (FACS) could represent a clear

solution for an independent annotation of the cells, paired multi-omics data with this type of annotation are lacking in the literature.

A limitation of Mowgli is that it does not offer a straightforward approach to defining the number of latent dimensions. The choice of number of latent dimensions ($k$) is inherently problem-dependent and several values of $k$ should be tested. This problem has been however extensively studied in NMF literature and classical tools like the cophenetic coefficient[75] or the elbow method[76] may guide the user's choice of $k$. At the same time, Mowgli displays relative robustness to changes in $k$ thus suggesting that small changes in $k$ will not affect its performance. In addition, OT is inherently expensive to compute compared to the Euclidean distance. But the entropic regularization of OT considered here is GPU-friendly, and GPU computations enable Mowgli to scale to the larger datasets presented in this article. The availability of GPUs is nowadays a standard in research centers and this will be further enhanced in the future, once larger single-cell datasets will be available. Regarding possible extensions of Mowgli, it would be interesting to deal with batch correction once integrating paired multi-omics data. Indeed, most recent large-scale paired multi-omics data are profiled in different centers thus creating batch correction issues.

# Methods

## Notations

Let us consider $n$ cells, measured across several modalities. Each modality $p$ has $m_p$ features (e.g. genes). Let us denote $\mathbf{A}^{(p)} \in \mathbb{R}_+^{m_p \times n}$ the dataset for modality $p$. Additionally, we impose each column of $\mathbf{A}^{(p)}$ to sum to 1, i.e. be a discrete probability distribution.

## Optimal transport

Optimal Transport (OT), as defined by Monge[32] and Kantorovich[77], aims at comparing two probability distributions $a$ and $b$ by computing the minimal cost of transporting one distribution to the other. In the discrete case, the classical OT distance, also known as the Wasserstein distance, between $m$-dimensional histograms $\mathbf{a} = (a_1, \ldots, a_m)$ and $\mathbf{b} = (b_1, \ldots, b_m)$ is defined as

$$\mathrm{OT}(\mathbf{a}, \mathbf{b}) = \min_{\mathbf{P} \in \Pi(\mathbf{a}, \mathbf{b})} \sum_{k,l} P_{k,l} C_{k,l} \tag{2}$$

where $\Pi(\mathbf{a}, \mathbf{b}) = \{\mathbf{P} \in \mathbb{R}_+^{m \times m} \text{ such that} \sum_l P_{k,l} = a_k \text{ and } \sum_k P_{k,l} = b_l\}$.

In this discrete case, the coupling $\mathbf{P} \in \Pi(\mathbf{a}, \mathbf{b})$ is a matrix that represents how the mass in the discrete probability distribution $\mathbf{a}$ is moved from one bin (e.g. gene) to another one in order to transform $\mathbf{a}$ into $\mathbf{b}$. In other words, $P_{k,l}$ is the amount of gene expression transported between genes $k$ and $l$ when transporting the gene expression profile of the cell $\mathbf{a}$ to the gene expression profile of the cell $\mathbf{b}$.

The ground cost $\mathbf{C} \in \mathbb{R}_+^{m \times m}$ is a pairwise distance matrix that encodes the penalty for transporting mass from one feature (e.g. gene) to another. Hence, $\mathbf{C}$ should be chosen in such a way that similar bins (e.g. genes) $k$ and $l$ have a low cost $C_{k,l}$. Indeed, this will favor transporting gene expression between similar genes. Here, for a certain omic $p$, we define $C$ in a data-driven way as the matrix of pairwise cosine distances between the features, i.e. the rows in our dataset $\mathbf{A}^{(p)}$. In other words, denoting $\mathbf{u}_k \in \mathbb{R}_+^n$ and $\mathbf{u}_l \in \mathbb{R}_+^n$ two rows in our dataset,

$$C_{k,l} = 1 - \frac{\langle \mathbf{u}_k, \mathbf{u}_l \rangle}{||\mathbf{u}_k||_2 ||\mathbf{u}_l||_2} \tag{3}$$

where $\langle \mathbf{x}, \mathbf{y} \rangle = \sum_i x_i y_i$ is the dot product. This choice of ground cost gave the best results in our previous work[33].

Due to the high dimensionality of single-cell data, we use an approximation of classical OT. The entropic regularization of OT, described in (Eq. 4) below is computed using the fast and GPU-enabled Sinkhorn algorithm[78].

$$\mathrm{OT}_\varepsilon(\mathbf{a}, \mathbf{b}) = \min_{\mathbf{P} \in \Pi(\mathbf{a}, \mathbf{b})} \sum_{k,l} P_{k,l} C_{k,l} - \varepsilon E(\mathbf{P}) \tag{4}$$

where the entropy E is defined as $E : \mathbf{X} \in \mathbb{R}_+^{m \times n} \mapsto -\sum_{k,l} X_{k,l} \log X_{k,l}$.

If $\varepsilon$ is set to zero, (Eq. 4) corresponds exactly to classical OT (Eq. 2). Increasing values of $\varepsilon$ correspond to a more diffused coupling $\mathbf{P}$. In previous work, we showed the entropic regularization of OT to improve similarity inference between single-cell omics profiles compared to classical notions of distance[33].

As explored in that work[33], entropic regularization is expected to control the systematic noise due to technical dropouts and to the stochasticity of gene expression at the single-cell level. In addition, more diffused couplings increase the exchange of mass between features. This enables OT to leverage the relationships between features (e.g. genes), motivating further its application to single-cell data.

## Mowgli

We aim to decompose each matrix $\mathbf{A}^{(p)}$ as the product of a matrix $\mathbf{H}^{(p)} \in \mathbb{R}_+^{m_p \times k}$ (the modality-specific dictionaries) and $\mathbf{W} \in \mathbb{R}_+^{k \times n}$ (the embeddings, shared across modalities). The integer $k$ is the number of dimensions of the latent space and should be small compared to the number of features. We use the entropic regularization of OT as a reconstruction loss to compare $\mathbf{H}^{(p)}\mathbf{W}$ to the reference data $\mathbf{A}^{(p)}$.

In addition, we require the columns of $\mathbf{H}^{(p)}\mathbf{W}$ to sum to 1, i.e. belong to the simplex. We thus impose that the columns of $\mathbf{H}^{(p)}$ sum to one, and that the columns of $\mathbf{W}$ sum to one. Following Rolet et al.[42], we use the entropy function E defined previously, with a value of $-\infty$ when columns do not sum to one.

Combining the reconstruction and the entropy terms yields the loss

$$\sum_p \left( \sum_j \mathrm{OT}_\varepsilon \left( \mathbf{H}^{(p)} \mathbf{w}_j, \mathbf{a}_j^{(p)} \right) - \rho_p E \left( \mathbf{H}^{(p)} \right) - \mu E(\mathbf{W}) \right). \tag{5}$$

Note that for the sake of readability, we write $\mathrm{OT}_\varepsilon$ for all $p$, but this loss actually depends on an omic-specific ground cost $\mathbf{C}^{(p)}$, which itself depends on $\mathbf{A}^{(p)}$ (see Eq. 3). The parameters $\rho_p$, control the sparsity of the columns of $\mathbf{H}^{(p)}$ and $\mathbf{W}$. In order to make these parameters more comparable across omics and datasets, we define

$$\rho_p = \frac{1}{\log m_p \times k} \tilde{\rho}_p \quad \text{and} \quad \mu = \frac{1}{\log k \times n} \tilde{\mu}. \tag{6}$$

We have set the values of $\tilde{\rho}_{rna} = 0.01$, $\tilde{\rho}_{adt} = 0.01$, $\tilde{\rho}_{atac} = 0.1$ and $\tilde{\mu} = 0.001$ since they performed best across datasets and metrics (silhouette score, ARI, purity score) (see Supplementary Figs 9, 10, 11). Regarding the choice of number of latent dimensions, for *Liu* and datasets derived from *Liu*, we run the method with 5 factors. For other datasets, we choose 50 factors. These parameters gave the best results overall (see Supplementary Fig. 12).

Similarly to Rolet et al.[42], we alternate between minimizing (Eq. 5) on $\mathbf{H}^{(p)}$ and $\mathbf{W}$. One can show that these smooth minimization problems on $\mathbf{H}^{(p)}$ and $\mathbf{W}$ are equivalent to the following smooth minimization problems on new dual variables $\mathbf{G}^{(p)}$. These problems can be solved using standard optimization methods, and the method of choice is L-BFGS, a limited-memory quasi-Newton method.

- *Optimizing* $\mathbf{H}^{(p)}$. We solve the following smooth minimization problem:

$$\min_{\mathbf{G}^{(p)}} \sum_j \left( \mathrm{OT}_\varepsilon^\star \left( \mathbf{g}_j^{(p)}, \mathbf{a}_j^{(p)} \right) \right) - \rho_p(-\mathrm{E})^\star \left( -\mathbf{G}^{(p)}\mathbf{W}^\top / \rho_p \right) \quad (7)$$

Then, we update the primal variable as follows:

$$\mathbf{H}^{(p)} = \mathrm{softmax}\left( -\mathbf{G}^{(p)}\mathbf{W}^\top / \rho_p \right) \quad (8)$$

The column-wise softmax is defined as:

$$\mathrm{softmax} : \mathbf{X} \in \mathbb{R}^{m \times n} \mapsto \frac{\exp\left(X_{i,j}\right)}{\sum_i \exp\left(X_{i,j}\right)} \quad (9)$$

- *Optimizing* $\mathbf{W}$. We solve the following smooth minimization problem:

$$\min_{\mathbf{G}^{(p)}} \sum_\rho \left( \sum_j \mathrm{OT}_\varepsilon^\star \left( \mathbf{g}_j^{(p)}, \mathbf{a}_j^{(p)} \right) \right) - d\mu(-\mathrm{E})^\star \left( -\frac{1}{d\mu} \sum_\rho \mathbf{H}^{(p)\top} \mathbf{G}^{(p)} \right) \quad (10)$$

Then, we update the primal variable as follows:

$$\mathbf{W} = \mathrm{softmax}\left( -\frac{1}{d\mu} \sum_\rho \mathbf{H}^{(p)\top} \mathbf{G}^{(p)} \right) \quad (11)$$

Here, $\mathrm{OT}_\varepsilon^\star$ and $(-\mathrm{E})^\star$ denote the Legendre duals of the $\mathrm{OT}_\varepsilon$ and $-\mathrm{E}$ functions, and their smooth closed form expressions are defined in Rolet et al.[42]

An important property of NMF enabling representation by parts is sparsity, but it is not enforced explicitly[79]. In classical NMF, the L1 norm can be leveraged to explicitly enforce sparsity[79,80]. In our setting, the simplex constraint renders the L1 penalty ineffective (because it is equal to 1), but we can leverage the entropic penalty of Rolet et al.[42] to control sparsity. Indeed, the coefficients $\rho_p$ and $\mu$ parametrize *softmax* functions, and hence control the sparsity of distributions. When setting $\rho_p$ (or $\mu$) to zero, the solutions are Dirac masses (extreme sparsity) while when setting it to $+\infty$ the solutions are uniform (no sparsity).

The code is implemented in Python and relies on PyTorch[81] for matrix operations on the GPU and on Muon[47] and Scanpy[46] to handle single-cell multimodal data. The compared running times of Mowgli and other methods is reported in Supplementary Table 2.

## MOFA+
We compare Mowgli to MOFA+[14], a variational inference method analogous to sparse PCA for multi-omic data. We use the R interface MOFA2 with default training parameters. MOFA+ provides a parameter drop_factor_threshold designed to keep only informative factors, but we found that in practice it removed important information. For example, the benchmark in Zuo and Chen[18] only kept one factor for MOFA+, which is not enough to represent cellular heterogeneity in the data. We thus choose to keep 5 factors for *Liu* and the datasets derived from *Liu*, and 15 factors for the other datasets. These parameters gave the best results overall (see Supplementary Fig. 13).

## Seurat v4
We compare Mowgli to Seurat v4[15] which uses Weighted Nearest Neighbors to integrate multi-omics data. We use the R interface Seurat with default parameters.

## Cobolt
We compare Mowgli to Cobolt[26], a deep neural network approach to integrate multi-omics data. We use the Python package cobolt with a learning rate of 0.001 and 200 epochs. As suggested in the documentation, we use raw counts as input for the RNA and ATAC modalities. For *Liu* and the datasets derived from *Liu*, we use 5 latent dimensions, and for the other datasets, we use 30 latent dimensions. These parameters gave the best results overall (see Supplementary Fig. 14).

## Multigrate
We compare Mowgli to Multigrate[27], another deep neural network approach to integrate multi-omics data. We use the Python package multigrate with 200 epochs. As suggested in the documentation, we use raw counts and a negative binomial loss for RNA and processed data with a mean squared error for ATAC and ADT. For *Liu* and the datasets derived from *Liu*, we use 15 latent dimensions, and for the other datasets, we use 50 latent dimensions. These parameters gave the best results overall (see Supplementary Fig. 15).

## Integrative NMF
We implemented a baseline NMF-based integration method by concatenating the features from the different omics and solving the optimization problem with positivity constraints:

$$\mathrm{argmin}_{\mathbf{H} \in \mathbb{R}_+^{m \times k}, \mathbf{W} \in \mathbb{R}_+^{k \times n}} ||\mathbf{A} - \mathbf{HW}||_2. \quad (12)$$

We implemented this approach using the TorchNMF package. For *Liu* and the datasets derived from *Liu*, we use 5 latent dimensions, and for the other datasets, we use 30 latent dimensions. These parameters gave the best results overall (see Supplementary Fig. 16).

Note that this is almost equivalent to intNMF[82] with $\theta = 1$, which minimizes instead $\sum_p ||\mathbf{A}^{(p)} - \mathbf{H}^{(p)}\mathbf{W}||_2$. However, on the considered datasets, the intNMF package was too slow to be able to include it in the benchmark.

## Choice of the number of latent dimensions for all methods
For each method described above, we selected the number of latent dimensions displaying the overall best performances across evaluation metrics (Silhouette score, ARI, purity score) and datasets. This analysis has been done considering Liu and the controlled settings separately from the other datasets (*PBMC 10X, OP Multiome, OP CITE, BM CITE*). Indeed, *Liu*, composed only of three cell lines, presents much less variation than the other real datasets.

## Data generation
**Mixed in RNA.** We simulate a dataset where one modality confuses two populations, while the other can separate them. To do so we replace the RNA profiles of all HCT cells with RNA profiles of random HeLa cells. ATAC profiles are left untouched.

**Mixed in both.** We simulate a dataset where the two modalities each confuse two cell populations, but separate two others. This makes the two omics complementary. To do so we replace the RNA profiles of all HCT cells with RNA profiles of random HeLa cells. Then, we replace the ATAC profiles of all K562 cells with ATAC profiles of random HeLa cells.

**Sparse.** We simulate high dropout noise by randomly replacing 50%, 70% or 90% of the values with zeros. Since the data is already 65% sparse, the final sparsity is 82%, 90% and 96%.

**Rare population.** We simulate the presence of a rare population by keeping only 10 randomly chosen HeLa cells.

## Data preprocessing

All preprocessing was performed using the Scanpy[46] and Muon[47] Python packages.

**RNA preprocessing.** Quality control filtering of cells was performed on the proportion of mitochondrial gene expression, the number of expressed genes, and the total number of counts (using Muon's filter_obs). Quality control filtering of genes was performed on the number of cells expressing the gene (using Muon's filter_var). Cells were normalized to sum to 10000 (using Scanpy's normalize_total), then log-transformed (using Scanpy's log1p). The top 2500 most variable genes (1500 for the *Liu* dataset) were selected for downstream analysis (using Scanpy's highly_variable_genes with flavor='seurat').

**ATAC preprocessing.** Quality control filtering of cells was performed on the number of open peaks and the total number of counts (using Muon's filter_obs). Quality control filtering of peaks was performed on the number of cells where the peak is open (using Muon's filter_var). In *Liu*, *TEA*, and *10X PBMC*, cells were normalized to sum to 10000 (using Scanpy's normalize_total), then log-transformed (using Scanpy's log1p). In *OP Multiome*, cells were normalized using TF-IDF (using Muon's tfidf) to follow the pre-processing chosen by its authors. The most variable peaks were selected for downstream analysis (using Scanpy's highly_variable_genes with flavor='seurat'). Due to differences in the data's distribution across datasets, we chose to keep 1500 peaks in *Liu*, 5,000 peaks in PBMC, and 15,000 peaks in *OP Multiome* and *TEA*.

**ADT preprocessing.** Since the number of proteins is small and the data is less noisy than RNA or ATAC, no quality control or feature selection was performed. The data was normalized by Center Log Ratio (using Muon's clr).

## Data analysis

**Gene set enrichment analysis (GSEA).** The gProfiler API[83] was used through Scanpy's enrich. Custom sources GO:CC, GO: MF, GO: BP, Azimuth, and ImmuneSigDB were retrieved from the Enrichr website[84]. Gene sets enriched with adjusted *p*-values under 0.05 (with Bonferroni correction) were selected for further analysis. To make genes comparable, we normalized rows of the matrix $H^{(rna)}$ to 1. The 150 top genes for every factor were then used as an unordered input to gProfiler.

**Motif enrichment analysis.** Signac[85] was used to perform Motif Enrichment Analysis, using the JASPAR2022 motif database[86]. To make peaks comparable, we normalized rows of the matrix $H^{(atac)}$ to 1. The 100 top peaks for every factor were used as input to Signac's Find-Motifs. The union of the top peaks across factors constitutes the background.

**Visualization.** To visualize the latent representation of cells in MOFA+, integrative NMF, and Mowgli's models, we computed kNN graphs (k = 20) with the euclidean distance between the cells' low-dimensional embeddings (using Scanpy's neighbors). We used these graphs to compute 2-D UMAP[37] projections (using Scanpy's umap). For Seurat v4, 2-D UMAP projections based on WNN graphs were performed using Seurat v4's function RunUMAP.

**Clustering.** For Mowgli, integrative NMF, and MOFA+, we clustered datasets using the Leiden algorithm[38] with varying resolutions (using Scanpy's leiden). Similarly to UMAP visualization, the inputs of the Leiden algorithm were the previously computed kNN graphs. For Seurat v4, Leiden clustering was performed using Seurat v4's function FindClusters.

## Evaluation metrics

**Silhouette score.** For each sample, the silhouette width is defined as $\frac{b-a}{\max(a,b)}$ where $a$ is the mean distance of the sample to other samples of the same cluster and $b$ is the mean distance of the samples to samples from the nearest cluster. The silhouette score is the mean of silhouette widths across samples. The silhouette score varies between −1 and 1. We used Scikit-learn's implementation silhouette_score[87].

**kNN purity score.** The kNN purity score measures the average proportion of a sample's nearest neighbors that share the sample's cluster annotation. It thus varies between 0 and 1.

**Adjusted rand index.** The Rand Index defines the similarity between a ground truth annotation and an experimental clustering. The ARI is then defined as

$$ARI = \frac{RI - \mathbb{E}(RI)}{\max(RI) - \mathbb{E}(RI)} \tag{13}$$

and varies between −1 and 1, with 0 representing a random clustering. We used Scikit-learn's implementation adjusted_rand_score. Since the simulated datasets are derived from three cell lines, there is no biological heterogeneity within each of the three groups and clustering at high resolution necessarily leads to overclustering. Figure 2 thus displays maximum ARI, which is achieved for low resolutions and is more informative than the ARI across resolutions. ARI across resolutions can be found in Supplementary Figs 9 to 16.

## Specificity

MOFA+ was applied with 15 factors, which are enough to represent the data (see Supplementary Fig. 13). Integrative NMF was applied with 30 factors. Mowgli was applied with 50 factors. For all three methods, a coarse Leiden clustering was applied (using Scanpy's leiden). For all three methods, each cluster was assigned a cell type based on the expression of the canonical gene and protein markers (see Supplementary Fig. 2). To confirm this annotation, Azimuth was run on the RNA signal of the dataset (using the Azimuth web tool and the PBMC reference). The agreement of the four independent annotations is confirmed in a Sankey diagram (see Supplementary Fig. 3). Dendritic cells are absent in our manual annotations because of the coarseness of the clustering. Likewise, the ADT signal (see Supplementary Fig. 7) informs us that there is a CD4-CD8- T cell population missed in all four annotations. For each factor and each cell type, we computed (i) the proportion of cells within that cell type with an absolute weight greater than $10^{-3}$ (ii) $a$, the mean weight for cells within that cell type (iii) $b$, the mean weight for cells outside of that cell type. For each cell type, we then defined a specificity score for factor $i$:

$$specificity_i = \frac{a_i - b_i}{\max_j a_j} \tag{14}$$

where $\max_j a_j$ is the maximum value of $a$ computed over all factors. The specificity score is thus bounded by 1. See Fig. 4C for a visualization of this information and see Supplementary Fig. 5 for results across hyperparameters μ and numbers of latent dimensions.

## Biological interpretation

We added stars in front of biologically interesting elements in Fig. 5C. The first resource we used was the Human Protein Atlas, from which we programmatically retrieved information about the top proteins and genes. We starred them if they were marked as specific to NK cells, Naive B cells, Memory B cells, or Memory CD8 T cells respectively. In addition, some genes or proteins were starred manually; we discuss those in the Results and refer to the relevant literature.

We starred gene sets if they matched the considered cell types, e.g. MHC II protein complex for B cells. To reduce the noise in the Immune gene sets, we only considered gene sets opposing subtypes of the broad cell type considered, e.g. NAIVE_VS_MEMORY_BCELL_UP.

We starred the TFs if they target one of the top 20 genes. For this, we retrieved TF-gene links from the Regulatory Circuits database[88] and considered the Natural Killer cells, CD19+B cells, and CD8+T cells networks.

**Reporting summary**

Further information on research design is available in the Nature Portfolio Reporting Summary linked to this article.

## Data availability

*Regulatory Circuits*. At the time of writing, the Regulatory Circuits website http://ww1.regulatorycircuits.org/ is down. We recovered the data from the mirror http://www2.unil.ch/cbg/regulatorycircuits/FANTOM5_individual_networks.tar. *PBMC*. We retrieve a 10X Genomics Multiome (RNA+ATAC) dataset with 9320 PBMCs. Data is available at https://www.10xgenomics.com/resources/datasets/pbmc-from-a-healthy-donor-granulocytes-removed-through-cell-sorting-10-k-1-standard-2-0-0. *Liu*. We retrieve a scCAT-seq (RNA+ATAC) dataset from Liu et al.[8] with 206 cells from three cancer cell lines (HCT116, HeLa-S3, K562). Data is available in the Supplementary Materials of the original publication. *Simulated data*. Controlled settings derived from cell lines data were generated from the *Liu* dataset and can be reproduced using the provided reproducibility code (see Code availability). *OP-Multiome*. We retrieve a Multiome (RNA+ATAC) dataset from the Open Problems challenge[52] and select only the first batch, which contains 6137 BMMCs. The GEO accession number is GSE194122 and the data is available at. *OP-CITE*. We retrieve a CITE-seq (RNA+ADT) dataset from the Open Problems challenge[52] and select only the first batch, which contains 4249 BMMCs. The GEO accession number is GSE194122 and the data is available at. *BM-CITE*. We retrieve a CITE-seq (RNA +ADT) dataset from Stuart et al.[25] with 29,803 BMMCs. The GEO accession number is GSE128639 and the data is available at. *PBMC TEA-seq*. We retrieve a recent TEA-seq (RNA+ATAC+ADT) dataset from Swanson et al.[7] with 7084 PBMCs. The GEO accession number is GSE158013 and the data is available at.

## Code availability

*Package*. The Python package for Mowgli is hosted at https://github.com/cantinilab/mowgli/[89]. It can be installed easily by running pip install mowgli. *Reproducibility*. Code to reproduce the experiments and figures is available at https://github.com/cantinilab/mowgli_reproducibility/.

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

## Acknowledgements

We would like to thank Frank Augé and Tommaso Andreani for the insightful scientific discussions on this project in the context of the Sanofi iTech Awards. This work was supported by the Sanofi iTech Awards. The project leading to this manuscript has received funding from the Agence Nationale de la Recherche (ANR) JCJC project scMO-mix and the French government under management of Agence Nationale de la Recherche as part of the 'Investissements d'avenir' program, reference ANR19-P3IA-0001 (PRAIRIE 3IA Institute). The work of G. Peyré was supported by the European Research Council (ERC project NORIA) and the French government under management of Agence Nationale de la Recherche as part of the 'Investissements d'avenir' program, reference ANR19-P3IA-0001 (PRAIRIE 3IA Institute). GPU computations were performed using HPC resources from GENCI-IDRIS (Grant 2022-AD011012079R2).

## Author contributions

G-J.H., G.P. and L.C. designed and planned the study. G-J.H. and L.C. wrote the paper. G.P. revised the manuscript. G-J.H. developed the tool and performed all the analyses. IM.D. participated in the data analysis.

## Competing interests

The authors declare no competing interests.
