## [Peer Review File · Nature Communications]

Paired single-cell multi-omics data integration with MowgliREVIEWER COMMENTS

Reviewer #1 (Remarks to the Author):

Review on the paper "Paired single-cell multi-omics data integration with Mowgli" by Huizing et al.

In this article, Huizing et al. present a novel approach to perform multi-omics single-cell integration using a method combining integrative non-negative matrix factorization with optimal transport, which the authors have described in a previous publication. The method uses a novel loss function based on optimal transport instead of the classical Frobenius error which is usually used in NMF optimization. The authors benchmark this method against state-of-the-art multi-omics integration methods such as MOFA+ and Seurat, using a standard iNMF as a baseline method. The comparison is performed over several simulated datasets intended to challenge several aspects of single-cell multi-omics (sparsity, confusion of datasets in one modality, etc...). They also benchmark the method using several paired multi-omics datasets and compare the method using different metrics. The tool is provided as a python package compatible with standard python-based single-cell frameworks. The tool is well documented in the repository which makes it an easy-to-use tool for multi-omics data integration.

The article a well-written and clearly presents the different comparisons and benchmarks, which are illustrated by well-designed figures. I particularly appreciate Figure 4c, which is a very nice way to display generic/specific factors. My main concern is that this method while representing a nice complementary method to others, does not represent a significant improvement compared with the other tested methods. The gain in the different metrics (ARI, silhouette scores) is modest compared with the other approaches. Besides, some important aspects are not addressed with enough detail in the paper, in particular the influence of certain parameters which are simply set to specific values in the paper without detailed explanations of their potential impact.

1. My main concern has to do with the very crucial question of the choice of a number of factors. This is only addressed in the conclusion with a rather cryptic statement that "Mowgli is fairly robust to changes in the number of latent dimensions thus suggesting that small changes in will not affect its performance." Supplementary figures 7,8,9 present some evaluation of the effect of the number of latent dimensions on the performances (btw: left/right panels in Fig S7 should be swapped to be compatible with fig S8 and S9). The variation in the silhouette score for Mowgli might appear to be modest, they are in the range of the putative increase in performance discussed in the main text compared with the other methods. In other words, a different choice in the number of latent dimensions might lead to other conclusions in the performances of the different methods investigated, and a different ranking.

2. Related to the previous points, it is not explained how the value of k was selected. This is not explained in the main manuscript and is only mentioned in the methods section where different values are quoted: 5/50 depending on the dataset (line 482), 5 and 30 factors for MOFA+ (l. 522), then 15 factors (MOFA+, l. 636) and 50 for Mowgli (l. 637). It is unclear how these values were selected. I find the statement that “[...] This problem has been extensively studied in NMF literature and users can rely on classical tools like cophenetic distance or elbow method” rather optimistic. I feel that in most analyses, several values are tested and tried, and results are compared across different values of k to verify the robustness. Have the authors done this (apart from the plots in Figure S7,S8,S9)? For example, how would the conclusions of Figure 4c change (or not...) if different values of k had been chosen? The fact that MOFA+ displays several signatures (for example for MAIT-T-cells) could also be explained by the fact that the number of factors was too high.

3. The authors claim that Mowgli allows to identification of rare cell populations. They present some evidence for this in Figure 5. For example, they describe the plasmacytoid DC as being associated to a specific factor of Mowgli (Factor 3, Figure 5b). This factor seems to be associated to CD304 (Fig 5b). However, this coincidence is only shown for the cluster of B-cells, while Factor 3 also appears in other clusters, for example monocytes (Fig S4). Hence the claim that Factor 3 characterizes a specific subtype of DC, based on the plot in Fig 5b seems a bit vague. The claim that MOFA+ does not show any such factor also does not seem to be backed by facts or plots. Maybe the authors could perform a more quantitative analysis of the association of certain factors with specific ADT markers, and show a better association for one method vs. the other? Showing UMPA plots with minor changes in color intensity is not suitable for such a claim. However, this analysis should again consider the potential influence in the number of factors selected (see previous comments).

4. I am surprised by the additional loss term included in the loss function, and the claim that this is to ensure the sparsity of the W/H matrices. As far as I understand, the formulation of NMF by Lee and Seung does naturally (i.e. without the need for an additional regularization term) ensure sparsity in the decomposition, which is one of the major advantages of NMF compared to other decomposition methods. So why does this formulation require this factor? This introduces additional hyperparameters which might have an influence on the outcome. These parameters are set in the methods section (l. 480) with, again, little explanation or evaluation of their impact.

Reviewer #2 (Remarks to the Author):

Huizing et al. presents a rather interesting idea of using OT to improve the penalty function of joint NMF. The manuscript is generally well-written, but there are still multiple issues to be addressed before it's ready for publication.

1. More recent Deep Learning-based methods such as Cobolt, MultiVI, Multigrade, should be incorporated in the benchmark.
2. Meanwhile, the authors stated that "Mowgli is fairly robust to changes in the number of latent dimensions" in the conclusion section, with an evaluation in Suppl. Fig. 7. Nevertheless, the effect of other key hyperparameters (such as ρ and μ) on model performance should also be systematically evaluated.
3. In the TEA-seq case study, the authors only compared against MOFA+, but not regular NMF, which is also known to be interpretable. Without such comparison, it would be difficult to assess the benefit of the proposed OT penalty in this scenario.
4. The fact that Mowgli factors are more specific than MOFA+ could also be attributed to the different hyperparameter settings, e.g., Mowgli is run with 50 latent factors, while MOFA+ only 15. The larger number of Mowgli factors could also contribute to higher factor specificity. Will the comparison change dramatically if Mowgli is also run with 15 factors, or if MOFA+ was run with 50 factors?
5. Furthermore, the hyperparameter μ is also directly connected to factor specificity. A more comprehensive comparison with varying factor numbers and hyperparameter μ would be helpful to clarify this.
6. According to Fig. 3B, Mowgli's performance on real datasets does not outperform state-of-the-art methods. The authors argued that this is because cell type labels in most of these datasets were computationally defined and thus potentially inaccurate. While this sounds reasonable, it would be more convincing if the authors could provide examples where the Mowgli derived cell representations/clustering are more biologically informative compared to other methods, e.g., by examining the expression of established marker genes.

Minor issues

1. The x-axis in the ARI column in Fig. 3B (supposedly Leiden resolution?) is not labeled. Also, the reporting of benchmark results is not consistent, as a similar ARI-resolution curve is not shown in Fig. 2B where only the maximum ARI is reported.
2. The authors mentioned that Mowgli requires GPU to run on larger datasets, suggesting that the algorithm can be computationally intensive. How much time does it take for Mowgli to run on the benchmarked datasets? How does it compare with the other methods?

Reviewer #3 (Remarks to the Author):

The authors of this paper introduce Mowgli, a new method that uses integrative non-negative matrix factorization and employs the Wasserstein metric, also known as earth mover's distance, to compute reconstruction loss of input matrices. Additionally, entropic regularization is applied for this purpose.

The authors aptly demonstrate the utility of Mowgli on generated and real datasets and delve into the biological interpretability of the method to show its usefulness. Overall, the paper is well-written and flows logically. The provided package is also easy to install, and documentation support is available on their website.

I have the following comments:

1. Silhouette scores are missing for Seurat for both simulated and real data (Figures 2 and 3). They are also missing from the Github code repo. They should be included for the completeness of comparison.
2. Looking at the repo `cantinilab/mowgli_reproducibility`, it appears that a different number of neighbors were used for Seurat and other methods for generating the UMAP plot. It was 20 for Seurat, and 25 for others. Ideally, it should be the same between different methods unless there's a good reason.
3. This one is potentially problematic. Line 271, the authors make an argument that "MOFA+ tends to associate multiple factors to the same cell type." However, MOFA+ was run with 15 factors compared to Mowgli which was run with 50 allowing factors to resolve better. If the number of factors is smaller, biological processes would tend to group together. The explanation for this choice is given in methods section line 636 — "MOFA+ was applied with 15 factors, which are enough to represent the data 637 (see Supp Figure 8). Mowgli was applied with 50 factors." However, supp fig 8 which is MOFA+'s clustering and embedding quality plots does not resolve this. It is possible that authors are trying to select the minimum no of factors based on the 3 metrics, viz., silhouette, ARI, and purity. But these metrics do not capture the full scope of the data. Even based on these metrics, with 50 factors MOFA+ performs equally well and better in some cases compared to 15 factors. Granted, the silhouette score is higher with 15 factors in all cases, but apart from Liu dataset but it is comparable to 50. I would suggest the authors perform this analysis with a higher number of factors to provide strong support to their claim.

Following minor comments:

1. It seems like the authors intended to create well-spaced 50%, 70%, and 90% sparse data for the simulated dataset (iv). However, it resulted in 82%, 90%, and 96% sparse data because of pre-existing dropouts. If the authors wish, they could complete their intended goal by taking into account the existing dropouts, unless they are more than 50%.
2. Given the resolution in the pdf, the axis labels in fig 4C weren't clear. Probably increasing the font size a bit and making it darker would help.
3. The readability of the methods section where optimal transport is explained is low for someone unfamiliar with the concept. For example, some notations such as i , and j aren't explicitly mentioned. Although, they may be apparent to some readers. The paper flows well in most sections, however, in the first half of page 18, it doesn't. Minor section rewriting could be helpful to readers.
4. The values for regularization parameters used in the analysis are different from the default values of ``MowgliModel``. The documentation on `readthedocs.io` doesn't provide a description of these parameters which is already present in Github. So, updated documentation would be helpful to users.

Furthermore, some guidance on how to best determine the parameters in the documentation would also help.

5. Typo in line 553 - 'ot'

6. Figs 2B and 3B are missing some of the axis labels or the labels aren't explained well. For example, k is the number of nearest neighbors but it is not mentioned in the figure or text. The x-axis label for ARI is missing as well.

7. Supp Fig 1. The title of the top left fig is missing the tool name, MOFA+.

8. Fig 4C. Apparently, the factor number is mentioned inside the top-colored bubble, but the text or the legend doesn't mention it.

Additional feedback:

1. Typo in Github repo here:

<https://github.com/cantinilab/Mowgli/blob/1593dd5baf2161851339a14573ba890f376c2d1d/mowgli/models.py#L350>

2. It is good to see that the authors give an estimated amount of time required in the Github repo. Making it more detailed such as providing what kind of machine and OS was used would be helpful as well.

Point-by-point reply to Reviewers

We thank the Reviewers for their constructive remarks which helped improve the quality of the work. Below the point-by-point reply

Reviewer 1

In this article, Huizing et al. present a novel approach to perform multi-omics single-cell integration using a method combining integrative non-negative matrix factorization with optimal transport, which the authors have described in a previous publication. The method uses a novel loss function based on optimal transport instead of the classical Froebenius error which is usually used in NMF optimization. The authors benchmark this method against state-of-the-art multi-omics integration methods such as MOFA+ and Seurat, using a standard iNMF as a baseline method. The comparison is performed over several simulated datasets intended to challenge several aspects of single-cell multi-omics (sparsity, confusion of datasets in one modality, etc...). They also benchmark the method using several paired multi-omics datasets and compare the method using different metrics. The tool is provided as a python package compatible with standard python-based single-cell frameworks. The tool is well documented in the repository which makes it an easy-to-use tool for multi-omics data integration.

The article a well-written and clearly presents the different comparisons and benchmarks, which are illustrated by well-designed figures. I particularly appreciate Figure 4c, which is a very nice way to display generic/specific factors. My main concern is that this method while representing a nice complementary method to others, does not represent a significant improvement compared with the other tested methods. The gain in the different metrics (ARI, silhouette scores) is modest compared with the other approaches. Besides, some important aspects are not addressed with enough detail in the paper, in particular the influence of certain parameters which are simply set to specific values in the paper without detailed explanations of their potential impact.

1. My main concern has to do with the very crucial question of the choice of a number of factors. This is only addressed in the conclusion with a rather cryptic statement that "Mowgli is fairly robust to changes in the number of latent dimensions thus suggesting that small changes in will not affect its performance." Supplementary figures 7,8,9 present some evaluation of the effect of the number of latent dimensions on the performances (btw: left/right panels in Fig S7 should be swapped to be compatible with fig S8 and S9). The variation in the silhouette score for Mowgli might appear to be modest, they are in the range of the putative increase in performance discussed in the main text compared with the other methods. In other words, a different choice in the number of latent dimensions might lead to other conclusions in the performances of the different methods investigated, and a different ranking.

Reply: We agree with the Reviewer that the robustness of Mowgli with respect to the number of latent dimensions was not sufficiently investigated in the previous version of the paper. We now compiled the additional Supplementary Figure 1, which complements the main Figures 2 and 3 by displaying evaluation metrics with the best-performing number of latent dimensions for each dataset and metric.

As reported in the updated manuscript, tuning the number of latent dimensions does not change our conclusions, thus supporting Mowgli's robustness. In controlled settings, varying the number of latent dimensions, Mowgli still outperforms the other methods. In real datasets, some variations in performance are observed, but they do not advantage any method over the others. We now better discuss this at the end of sections 2 and 3 of results. In addition, we relaxed the sentence on Mowgli's robustness in conclusions:

"At the same time, Mowgli displays relative robustness to changes in k thus suggesting that small changes in k will not affect its performance."

2. Related to the previous points, it is not explained how the value of k was selected. This is not explained in the main manuscript and is only mentioned in the methods section where different values are quoted: 5/50 depending on the dataset (line 482), 5 and 30 factors for MOFA+ (l. 522), then 15 factors (MOFA+, l. 636) and 50 for Mowgli (l. 637). It is unclear how these values were selected.

Reply: In the updated manuscript, we explain our choice of the number of latent factors in the sections of Methods dedicated to the different tools. Shortly, for each method described above, we selected the number of latent dimensions displaying overall best performances across evaluation metrics (Silhouette score, ARI, purity score) and datasets. This analysis has been done considering Liu and the controlled settings separately from the other datasets (*PBMC 10X*, *OP Multiome*, *OP CITE*, *BM CITE*). Indeed, *Liu*, composed only of three cell lines, presents much less variation than the other real datasets. In addition, variations of performances across latent dimensions are now investigated in the new Supplementary Figure 1, as discussed above.

We also realized that the optimal latent dimensions value for MOFA+ reported in the text of the methods line 522 was the wrong one. We reported in the text a value of 30, while the correct one used for the figures was 15. We corrected the sentence in the updated manuscript.

I find the statement that "[...] This problem has been extensively studied in NMF literature and users can rely on classical tools like cophenetic distance or elbow method" rather optimistic.

Reply: We updated the manuscript with a more nuanced statement:

"The choice of number of latent dimensions (k) is inherently problem-dependent and several values of k should be tested. This problem has been however extensively studied in NMF literature and classical tools like the cophenetic coefficient⁷⁵ or the elbow method⁷⁶ may guide the user's choice of k . At the same time, Mowgli displays relative robustness to changes in k thus suggesting that small changes in k will not affect its performance."

I feel that in most analyses, several values are tested and tried, and results are compared across different values of k to verify the robustness. Have the authors done this (apart from the plots in Figure S7,S8,S9)? For example, how would the conclusions of Figure 4c change (or not...) if different values of k had been chosen? The fact that MOFA+ displays several signatures (for example for MAIT-T-cells) could also be explained by the fact that the number of factors was too high.

Reply: Variations of performances in cell embedding across latent dimensions are now investigated in the new Supplementary Figure 1, as discussed above. In addition, we now tested also the robustness of factors specificity related to Figure 4c.

All this is summarized in Supplementary Figure 5 complementing Figure 4c. Supplementary Figure 5A displays the specificity of Mowgli's factors with 15, 30, and 50 latent dimensions and shows that Mowgli's specificity is robust to the number of latent dimensions. Supplementary Figure 5C displays the specificity of MOFA+'s factors with 15, 30, and 50 latent dimensions. Increasing the number of factors for MOFA+ yields no additional cell-type-specific factors. Of note, upon request of Reviewer 2, we also added to this comparison integrative NMF for different latent dimensions and variations in Mowgli with respect to the parameter μ .

We now updated the Results section 4 at the end with:

“As shown in Supplementary Figure 5, our conclusions regarding the specificity of Mowgli's factors are robust to the choice of hyperparameters”

3. The authors claim that Mowgli allows to identification of rare cell populations. They present some evidence for this in Figure 5. For example, they describe the plasmacytoid DC as being associated to a specific factor of Mowgli (Factor 3, Figure 5b). This factor seems to be associated to CD304 (Fig 5b). However, this coincidence is only shown for the cluster of B-cells, while Factor 3 also appears in other clusters, for example monocytes (Fig S4). Hence the claim that Factor 3 characterizes a specific subtype of DC, based on the plot in Fig 5b seems a bit vague. The claim that MOFA+ does not show any such factor also does not seem to be backed by facts or plots. Maybe the authors could perform a more quantitative analysis of the association of certain factors with specific ADT markers, and show a better association for one method vs. the other? Showing UMPA plots with minor changes in color intensity is not suitable for such a claim. However, this analysis should again consider the potential influence in the number of factors selected (see previous comments).

Reply: We soften the claim regarding MOFA+ factors in the results:

“On the contrary, MOFA+'s factors most closely associated with the other immune subpopulations identified by Mowgli have less clear patterns (see Supplementary Text 2)”

At the same time, we agree with the Reviewer that more quantitative information was required. We thus now added Supplementary Text 2 reporting Mowgli's and MOFA+ factors most closely associated with certain markers (based on the position of the markers in their loadings), the comparison of the weights of these factors with their associated marker, and the positioning in the UMAP of the marker protein quantification vs factor activity. As shown in

Supplementary Text 2, Mowgli's factors are more closely associated with these markers than MOFA+'s factors in terms of similarity of distribution across cells of the markers and factors weights.

Regarding plasmacytoid DCs (pDCs), we agree with the Reviewer that factor 3 of Mowgli is associated with the ADT marker CD304, more than with plasmacytoid DCs as visible in Supplementary Text 2. Indeed, as noted by the Reviewer, both Factor 3 and the marker CD304 are present in the coarse B-cell cluster and in the coarse Monocyte cluster. This is in agreement with biology, as both Monocytes and pDCs express CD304. But CD304+ cells from the coarse Monocyte cluster express the CD14 marker, unlike CD304+ cells of the coarse B-cell cluster (see Supplementary Text 2). The CD304+CD14- subpopulation showcased in Figure 5B is thus coherent with plasmacytoid DC as described in [Matthew et al.].

We now mention the new Supplementary Text 2 in Results section 5 for a more quantitative association between factors and markers.

[Matthew et al.] Collin, Matthew, Naomi McGovern, and Muzlifah Haniffa. "Human dendritic cell subsets." *Immunology* 140.1 (2013): 22-30.

4. I am surprised by the additional loss term included in the loss function, and the claim that this is to ensure the sparsity of the W/H matrices. As far as I understand, the formulation of NMF by Lee and Seung does naturally (i.e. without the need for an additional regularization term) ensure sparsity in the decomposition, which is one of the major advantages of NMF compared to other decomposition methods. So why does this formulation require this factor?

Reply: We now modified Results section 1 to clarify that the formulation of NMF by [Lee and Seung] naturally induces sparsity:

"These terms ensure that the loadings and embeddings are positive distributions and they control their sparsity (see Methods), a crucial feature to further enhance the known NMF's "representation by parts" and sparsity properties."

But as explained in [Hoyer] and [Peharz and Pernkopf], NMF sparsity is a byproduct of nonnegativity, and NMF offers no parameter to control how sparse the representations are. Additional terms encouraging zero entries have thus been explored, such as the L1 norm [Hoyer] [Le Roux et al.] or directly the number of non-zero entries [Peharz and Pernkopf]. Note however that the setting we consider (Wasserstein NMF) is different from classical NMF because it includes a simplex (unit sum) constraint. This simplex constraint makes for instance the L1 penalty ineffective (because the L1 penalty is equal to 1).

We thus decided to use the entropic penalty proposed by [Rolet et al.] to control the sharpness of solutions. Although [Rolet et al.] do not use this penalty explicitly to control sparsity, we explain in Methods that when setting ρ (or μ) to zero, the solutions are Dirac masses (extreme sparsity) while when setting it to $+\infty$ the solutions are uniform (no sparsity).

To further clarify this, we also modified methods section Mowgli adding:

“An important property of NMF enabling “representation by parts” is sparsity, but it is not enforced explicitly⁷⁹. In classical NMF, the L1 norm can be leveraged to explicitly enforce sparsity^{79,80}. In our setting, the simplex constraint renders the L1 penalty ineffective (because it is equal to 1), but we can leverage the entropic penalty of Rolet et al.⁴² to control sparsity. Indeed, the coefficients ρ_p and μ parametrize *softmax* functions, and hence control the sparsity of distributions. When setting ρ_p (or μ) to zero, the solutions are Dirac masses (extreme sparsity) while when setting it to $+\infty$ the solutions are uniform (no sparsity).”

References:

- [Lee and Seung] Lee, Daniel D., and H. Sebastian Seung. "Learning the parts of objects by non-negative matrix factorization." *Nature* 401.6755 (1999): 788-791.
- [Hoyer] Hoyer, Patrik O. "Non-negative matrix factorization with sparseness constraints." *Journal of machine learning research* 5.9 (2004).
- [Peharz and Pernkopf] Peharz, Robert, and Franz Pernkopf. "Sparse nonnegative matrix factorization with ℓ_0 -constraints." *Neurocomputing* 80 (2012): 38-46.
- [Le Roux et al.] Le Roux, Jonathan, Felix J. Weninger, and John R. Hershey. "Sparse NMF—half-baked or well done?." *Mitsubishi Electric Research Labs (MERL), Cambridge, MA, USA, Tech. Rep., no. TR2015-023* 11 (2015): 13-15.
- [Rolet et al.] Rolet, Antoine, Marco Cuturi, and Gabriel Peyré. "Fast dictionary learning with a smoothed Wasserstein loss." *Artificial Intelligence and Statistics*. PMLR, 2016.

This introduces additional hyperparameters which might have an influence on the outcome. These parameters are set in the methods section (l. 480) with, again, little explanation or evaluation of their impact.

Reply: All parameters in Mowgli (μ , ρ_{rna} , ρ_{atac} , ρ_{adt} and latent dimensions) have been chosen based on those providing the best performances across datasets and metrics (silhouette score, ARI, purity score). We now better clarify this point in the Methods section referring to Mowgli:

“We have set the values of $\rho_{rna} = 0.01$, $\rho_{adt} = 0.01$, $\rho_{atac} = 0.1$, and $\mu = 0.001$ since they performed best across datasets and metrics (silhouette score, ARI, purity score) (See Supplementary Figures 9, 10, 11). Regarding the choice of number of latent dimensions, for Liu and datasets derived from Liu, we run the method with 5 factors. For other datasets, we choose 50 factors. These parameters gave the best results overall (see Supplementary Figure 12).”

As suggested by the Reviewer, we performed a systematic evaluation of the influence of hyperparameters (μ , ρ_{rna} , ρ_{atac} , ρ_{adt} and latent dimensions) and we provided the additional Supplementary Figures 9, 10, 11, 12. This extensive benchmark validates the default values described in the manuscript and suggests that Mowgli is robust to changes in these hyperparameters.

In addition, Supplementary Figure 5 evaluates the effect of μ on the specificity of Mowgli's factors, showing that the default value $\mu=0.001$ yields the best results.

Reviewer 2

Huizing et al. presents a rather interesting idea of using OT to improve the penalty function of joint NMF. The manuscript is generally well-written, but there are still multiple issues to be addressed before it's ready for publication.

Major issues

1. More recent Deep Learning-based methods such as Cobolt, MultiVI, Multigrade, should be incorporated in the benchmark.

Reply: We thank the Reviewer for this valuable remark that concretely impacted the value of the work. We added Multigrade and Cobolt to our benchmark and updated the manuscript accordingly (see new Figures 2 and 3). As discussed at the end of Results section 1, we here compared Mowgli with respect to state-of-the-art methods flexibly dealing with whatever combination of data. Since MultiVI operates similarly to Multigrade and Cobolt but only deals with 10X Multiome data (scRNA + scATAC data), we thus did not add it to the benchmark.

In controlled settings, Mowgli outperforms Multigrade and Cobolt. In particular, in the *Mixed in RNA* and *Mixed in both* datasets, Cobolt does not manage to identify the three populations of cells. In real data, Multigrade and Cobolt show results comparable to other methods. In addition, for sparsity 96% both Multigrade and Cobolt mix the three populations of cells, while Mowgli correctly separates them.

Finally, we added Supplementary Figures 14 and 15 displaying evaluation metrics for Cobolt and Multigrade across numbers of latent dimensions. The considerations made above are not affected by the number of latent dimensions.

We now modified the text in Results sections 2 and 3 mentioning Multigrade and Cobolt performance. In addition, we added their description in the Methods section.

2. Meanwhile, the authors stated that "Mowgli is fairly robust to changes in the number of latent dimensions" in the conclusion section, with an evaluation in Suppl. Fig. 7. Nevertheless, the effect of other key hyperparameters (such as ρ and μ) on model performance should also be systematically evaluated.

Reply: We now soften the sentence in the Conclusions with:

"At the same time, Mowgli displays relative robustness to changes in k thus suggesting that small changes in k will not affect its performance."

In addition, we systematically evaluated the influence of hyperparameters μ , ρ_{ma} , ρ_{atac} , ρ_{adt} , and the number of latent dimensions), reporting results in Supplementary Figures 9, 10, 11, 12. Our benchmark validates the default values described in the manuscript.

In addition, we provide an evaluation of the effect of μ on the specificity of Mowgli's factors in our response to point 5.

3. In the TEA-seq case study, the authors only compared against MOFA+, but not regular NMF, which is also known to be interpretable. Without such comparison, it would be difficult to assess the benefit of the proposed OT penalty in this scenario.

Reply: As suggested by the Reviewer, we updated the Results and Figure 4C with specificity evaluation for integrative NMF. Interestingly, Mowgli outperforms both MOFA+ and NMF in terms of specificity, further strengthening our argument.

We also provide an evaluation of NMF's specificity for different numbers of latent dimensions in Supplementary Figure 5, which shows that our conclusions are robust to the number of latent factors considered.

We now modified the text of Results section 4 discussing these aspects, plus the methods section on specificity to give all details of how this analysis has been performed with integrative NMF.

4. The fact that Mowgli factors are more specific than MOFA+ could also be attributed to the different hyperparameter settings, e.g., Mowgli is run with 50 latent factors, while MOFA+ only 15. The larger number of Mowgli factors could also contribute to higher factor specificity. Will the comparison change dramatically if Mowgli is also run with 15 factors, or if MOFA+ was run with 50 factors?

Reply: This point is addressed in the response to Reviewer 1. The additional Supplementary Figure 1 evaluates specificity for Mowgli, MOFA+, and integrative NMF across the number of latent dimensions. Our conclusions are robust to the number of latent dimensions. In particular, the requested MOFA+ with 50 factors does not provide better performances than Mowgli.

Importantly, MOFA+ factors are ranked based on variance explained and are thus of decreasing importance. As in a PCA, the first 15 factors of MOFA+ when run with $k=50$ will be the same as when run with $k=15$.

5. Furthermore, the hyperparameter μ is also directly connected to factor specificity. A more comprehensive comparison with varying factor numbers and hyperparameter μ would be helpful to clarify this.

Reply: We provide an evaluation of Mowgli's specificity for different values of the hyperparameter μ in the additional Supplementary Figure 5. While Mowgli's specificity is robust to the choice of the parameter μ , the default value $\mu=0.001$ yields the best results.

6. According to Fig. 3B, Mowgli's performance on real datasets does not outperform state-of-the-art methods. The authors argued that this is because cell type labels in most of these datasets were computationally defined and thus potentially inaccurate. While this sounds reasonable, it would be more convincing if the authors could provide examples where the Mowgli derived cell representations/clustering are more biologically informative compared to other methods, e.g., by examining the expression of established marker genes.

Reply: In order to quantitatively evaluate Mowgli's biological insights when they conflict with the annotation, we performed an in-depth analysis outlined in Supplementary Text 1. We now refer to this new Supplementary Text in the Results section 3.

Supplementary Text 1 explains two examples where the cell type annotation negatively affects our benchmark. First, we identify an inconsistency in the annotation of one batch in the *OP CITE* dataset regarding the CD8+TIGIT+CD45RA+ and CD8+TIGIT+CD45RO+ cell types. Second, we show that the granularity of the annotation in *BM CITE* limits the measured performance of Mowgli in the CD8 Effector_1 and CD8 Memory_2 cell types.

For all details on this part refer to Supplementary Text 1.

Minor issues

1. The x-axis in the ARI column in Fig. 3B (supposedly Leiden resolution?) is not labeled. Also, the reporting of benchmark results is not consistent, as a similar ARI-resolution curve is not shown in Fig. 2B where only the maximum ARI is reported.

Reply: The labeling issue is fixed in the updated version of Figure 3. We chose to show the maximum ARI in Figure 2B, as it is more informative because of the low number of expected clusters (three). Indeed, increasing subclustering leads to decreasing ARI even if the method correctly separates the three signals, as is visible in Figure 3B for *Liu*. However, the raw ARI curves can be observed in Supplementary Figures 9 to 16.

2. The authors mentioned that Mowgli requires GPU to run on larger datasets, suggesting that the algorithm can be computationally intensive. How much time does it take for Mowgli to run on the benchmarked datasets? How does it compare with the other methods?

Reply: We added a new Supplementary Table 2 comparing the runtimes of the different methods. Mowgli's computational cost is generally higher than the other benchmarked methods but remains reasonable in the context of the analysis of large, high-dimensional datasets. Importantly, Mowgli's speed is impacted by the number of features, because it considers interactions between them. This explains why Mowgli runs faster on *BM CITE* than *OP Multiome* despite having five times as many cells.

Reviewer 3

The authors of this paper introduce Mowgli, a new method that uses integrative non-negative matrix factorization and employs the Wasserstein metric, also known as earth mover's distance, to compute reconstruction loss of input matrices. Additionally, entropic regularization is applied for this purpose. The authors aptly demonstrate the utility of Mowgli on generated and real datasets and delve into the biological interpretability of the method to show its usefulness. Overall, the paper is well-written and flows logically. The provided package is also easy to install, and documentation support is available on their website.

Major comments

1. Silhouette scores are missing for Seurat for both simulated and real data (Figures 2 and 3). They are also missing from the Github code repo. They should be included for the completeness of comparison.

Reply: The absence of a silhouette score for Seurat v4 is intentional. The silhouette score requires an embedding of the data, which Seurat v4 does not provide, unlike the other methods in this benchmark. Indeed, Seurat v4 constructs a WNN graph based on a cell's neighbors across modalities, and this graph is used for clustering and visualization. We clarified this in the updated manuscript Results section 2 and We now mentioned "not applicable for Seurat" in Figure 2 and Figure 3 to make it clear that this absence is intentional.

2. Looking at the repo cantinilab/mowgli_reproducibility, it appears that a different number of neighbors were used for Seurat and other methods for generating the UMAP plot. It was 20 for Seurat, and 25 for others. Ideally, it should be the same between different methods unless there's a good reason.

Reply: We thank the Reviewer for this observation that we had not realized. There is indeed a difference in parameters. However, it is not between Seurat and other methods but rather between the UMAPs of the TEA-seq dataset and the UMAPs of the other datasets. We set the parameter to 20 (Seurat's default) everywhere for homogeneity. For this reason, UMAP plots in Figure 4, Figure 5 and Supplementary Figures 4,6,7,8 have been updated.

3. This one is potentially problematic. Line 271, the authors make an argument that "MOFA+ tends to associate multiple factors to the same cell type." However, MOFA+ was run with 15 factors compared to Mowgli which was run with 50 allowing factors to resolve better. If the number of factors is smaller, biological processes would tend to group together. The explanation for this choice is given in methods section line 636 — "MOFA+ was applied with 15 factors, which are enough to represent the data 637 (see Supp Figure 8). Mowgli was applied with 50 factors." However, supp fig 8 which is MOFA+'s clustering and embedding quality plots does not resolve this. It is possible that authors are trying to select the minimum no of factors based on the 3 metrics, viz., silhouette, ARI, and purity. But these metrics do not capture the full scope of the data. Even based on these metrics, with 50 factors MOFA+ performs equally well and better in some cases compared to 15 factors. Granted, the silhouette score is higher with 15 factors in all cases, but apart from Liu dataset but it is comparable to 50. I would suggest the authors perform this analysis with a higher number of factors to provide strong support to their claim.

Reply: We addressed this point in Supplementary Figure 5, which displays the specificity of MOFA+ and Mowgli's factors with 15, 30, and 50 latent dimensions and shows that our conclusions are robust to the number of latent dimensions. In particular, the requested MOFA+ with 50 factors does not provide better performances than Mowgli.

Importantly, while the Reviewer's comment about biological processes grouping together for smaller numbers of factors is correct in general, this is not the case for MOFA+. Indeed, as

in PCA, its factors are ranked based on their variance explained, and thus the biological signal in each individual factor does not depend on the number of factors.

Minor comments

1. It seems like the authors intended to create well-spaced 50%, 70%, and 90% sparse data for the simulated dataset (iv). However, it resulted in 82%, 90%, and 96% sparse data because of pre-existing dropouts. If the authors wish, they could complete their intended goal by taking into account the existing dropouts, unless they are more than 50%.

Reply: The preexisting dropouts are 65%. Our intention with the 50%, 70%, and 90% values in the processing code was to create increasingly sparse data while remaining in the range of realistic data: the 10X PBMC data is 90% sparse after preprocessing. We clarified this in Methods.

2. Given the resolution in the pdf, the axis labels in fig 4C weren't clear. Probably increasing the font size a bit and making it darker would help.

Reply: We increased the font size and the output resolution of the figures.

3. The readability of the methods section where optimal transport is explained is low for someone unfamiliar with the concept. For example, some notations such as i , and j aren't explicitly mentioned. Although, they may be apparent to some readers. The paper flows well in most sections, however, in the first half of page 18, it doesn't. Minor section rewriting could be helpful to readers.

Reply: As suggested by the Reviewer we explicited unclear notations and reworked the introduction to optimal transport.

4. The values for regularization parameters used in the analysis are different from the default values of `MowgliModel`. The documentation on readthedocs.io doesn't provide a description of these parameters which is already present in Github. So, updated documentation would be helpful to users. Furthermore, some guidance on how to best determine the parameters in the documentation would also help.

Reply: We thank the Reviewer for pointing this out and in general for helping to improve the quality of the Mowgli repository. We have updated the default values for RNA, ATAC, and ADT in the package and updated the function's documentation with more guidance.

5. Typo in line 553 - 'ot'
6. Figs 2B and 3B are missing some of the axis labels or the labels aren't explained well. For example, k is the number of nearest neighbors but it is not mentioned in the figure or text. The x-axis label for ARI is missing as well.
7. Supp Fig 1. The title of the top left fig is missing the tool name, MOFA+.
8. Fig 4C. Apparently, the factor number is mentioned inside the top-colored bubble, but the text or the legend doesn't mention it.

Reply: The typos and legend adjustments pointed out above were fixed in the updated version of the manuscript

Additional feedback

1. Typo in Github repo here:

<https://github.com/cantinilab/Mowgli/blob/1593dd5baf2161851339a14573ba890f376c2d1d/mowgli/models.py#L350>

Reply: The typo has been fixed in the repository.

2. It is good to see that the authors give an estimated amount of time required in the Github repo. Making it more detailed such as providing what kind of machine and OS was used would be helpful as well.

Reply: The readme has been updated with the relevant information. In addition, we provided more extensive runtime information in Supplementary Table 2.

REVIEWER COMMENTS

Reviewer #1 (Remarks to the Author):

I thank the authors of the paper for their extensive work in addressing my previous comments. I feel that most of my concerns have been addressed, however, I still would like to point out one issue:

1. Contrary to what the authors are stating, the new supplementary figure 1 does not address the question of the variability of results depending on the number of latent factors. It merely shows a more detailed plot compared to Figure 2, in which the optimal number of selected dimensions is indicated. I do not see in this particular plot any comparison, for one single method, across different values of k ! To be honest, I do not really see the added value of Supplementary Figure 1, except the presentation of the ARI results for the simulated datasets, which displays the dependency on the resolution parameter in the supplementary figure, while it does not in the main figure 2. I am not sure what the little diagram in Figure S1 (bottom right) is supposed to display though.

So I would strongly suggest to replace the "Maximum ARI score" plot in Figure 2B with the ARI Score plot of Figure S1 for the simulated datasets, to have a consistency between Figure 2 and Figure 3, and dropping Figure S1.

Thanks for the clarification on the sparsity question of NMF, I was not aware of these other aspects detailed in the rebuttal!

Reviewer #2 (Remarks to the Author):

I'd appreciate the authors' positive attitude toward addressing the reviewer's comments. Most of the comments I made have been well addressed except for one:

While I appreciate the authors' additional assessments on the limitation of imperfect "ground truth" annotation, I still don't quite understand that how such limitation could be connected to the degraded performance of Mowgli when being compared to other tools: would such defect/data glitch fail every benchmarked algorithms fairly?

Reviewer #3 (Remarks to the Author):

The authors have done a good job of addressing my previous concerns in their answers and newly added information and figures. Here are a few minor comments I have to help improve the manuscript and its utility to a wider group of readers:

1. As I mentioned earlier in my minor comment 4., the documentation of Mowgli on readthedocs.io (<https://mowgli.readthedocs.io/en/latest/models.html>) is missing a description of parameters such as `h_regularization`. The description is already present in Github. Also, documentation for mowgli.score api is missing as well. So, if it could be included in the documentation, it'll help a wider group of users.
2. Line 682: In the specificity formula, j , although obvious, is not defined.

We thank the Reviewers for this second round of constructive remarks which helped further improve the quality of the work. Below is the point-by-point reply to their comments.

Reviewer 1

I thank the authors of the paper for their extensive work in addressing my previous comments. I feel that most of my concerns have been addressed, however, I still would like to point out one issue:

1. Contrary to what the authors are stating, the new supplementary figure 1 does not address the question of the variability of results depending on the number of latent factors. It merely shows a more detailed plot compared to Figure 2, in which the optimal number of selected dimensions is indicated. I do not see in this particular plot any comparison, for one single method, across different values of k ! To be honest, I do not really see the added value of Supplementary Figure 1, except the presentation of the ARI results for the simulated datasets, which displays the dependency on the resolution parameter in the supplementary figure, while it does not in the main figure 2. I am not sure what the little diagram in Figure S1 (bottom right) is supposed to display though. So I would strongly suggest to replace the “Maximum ARI score” plot in Figure 2B with the ARI Score plot of Figure S1 for the simulated datasets, to have a consistency between Figure 2 and Figure 3, and dropping Figure S1.

Reply: We are sorry if the explanation in the reply to the Reviewer and caption of Supplementary Figure 1 were not clear and we now realize that its contents might have been misunderstood.

We proposed Supplementary Figure 1 in response to the following comment of this Reviewer: *“In other words, a different choice in the number of latent dimensions might lead to other conclusions in the performances of the different methods investigated, and a different ranking.”*. To address this comment in the previous revision we provided Supplementary Figure 1 showing for each method and metric the performances obtained for the best-performant number of latent dimensions. If a method has best performances in Supplementary Figure 1, then it means that the other methods are less performant no matter the number of latent dimensions chosen. On the opposite, the plots in Figure 2 and 3 are different. Indeed, in Figure 2 and 3, as explained in the methods section of the paper, for each method and dataset the evaluation metrics are computed fixing the same value of latent dimensions. We made this choice in Figures 2,3 as the best performing number of latent dimensions might vary from metric to metric.

We now addressed the request of the Reviewer and updated Supplementary Figure 1 by compiling all values of the number of latent dimensions in the same plots. To improve readability in the plots for ARI and purity score, we highlighted the best-performing experiment for each combination of metric, method, and dataset. These highlighted experiments correspond to what was in the previous version of Supplementary Figure 1. In addition, as previously commented from Supplementary Figure 1, this does not change our conclusions, thus supporting Mowgli's robustness. In controlled settings, varying the number of latent

dimensions, Mowgli still outperforms the other methods. In real datasets, some variations in performance are observed, but they do not advantage any method over the others.

Finally, we chose to keep the maximum ARI in place of ARI across resolutions in Figure 2, as simulated data are composed of 3 clusters and thus high resolution results are not pertinent for performance evaluation.

We now updated the Methods by adding the following section to the paragraph referring to ARI:

“Since the simulated datasets are derived from three cell lines, there is no biological heterogeneity within each of the three groups and clustering at high resolution necessarily leads to overclustering. Figure 2 thus displays *maximum* ARI, which is achieved for low resolutions and is more informative than the ARI across resolutions. ARI across resolutions can be found in Supplementary Figures 9 to 16.”

Thanks for the clarification on the sparsity question of NMF, I was not aware of these other aspects detailed in the rebuttal!

Reviewer 2

I'd appreciate the authors' positive attitude toward addressing the reviewer's comments. Most of the comments I made have been well addressed except for one:

While I appreciate the authors' additional assessments on the limitation of imperfect “ground truth” annotation, I still don't quite understand that how such limitation could be connected to the degraded performance of Mowgli when being compared to other tools: would such defect/data glitch fail every benchmarked algorithms fairly?

Reply: We did not mean that an imperfect “ground truth” annotation would degrade only Mowgli's performance, but more in general, that it would affect all methods. For this reason, benchmarking should probably be complemented by an in-depth biological analysis of results. To make this point clearer we now modified the sentence in the middle of section 3 of Results with:

“On the contrary, the annotations of the other datasets were computationally derived, thus affecting this benchmark for all methods. Supplementary Text 1 illustrates this on specific subpopulations of CD8 T-cells in the *BM CITE* and *OP CITE* datasets.”

In addition, to better understand the impact of imperfect ground truth annotation on evaluation across methods, we updated Supplementary Text 1 with evaluation metrics in CD8+ T cells.

In the *OP CITE* dataset, we updated the annotation by fusing the CD8+ TIGIT+ CD45RA+ and CD8+ TIGIT+ CD45RO+ subtypes (see the new panel I). This does not affect the evaluation in the same way for all methods (see the new panels J, K, L). For instance, in terms of silhouette score, Mowgli, NMF, and Cobolt experience an improvement with the updated annotation but MOFA+ and Multigrade experience a decrease in performance.

In the *BM CITE* dataset, we split CD8+ T cell subtypes according to the CD57+/CD57- phenotypes instead of the CD45RA+/CD45RO+ phenotypes as in the original annotation (see the new panel G). This alternative way to split the same cells does not affect the evaluation in the same way for all methods (see the new panels H, I, J). For instance, the silhouette score of Mowgli, NMF, and Cobolt increases, while the silhouette score of MOFA+ and Multigrade decreases. It also differs across metrics, since the purity score of Mowgli decreases.

These additional experiments show that noisy annotation and heterogeneity within the annotation have an uneven impact on measured performance across methods.

Reviewer 3

The authors have done a good job of addressing my previous concerns in their answers and newly added information and figures. Here are a few minor comments I have to help improve the manuscript and its utility to a wider group of readers:

1. As I mentioned earlier in my minor comment 4., the documentation of Mowgli on [readthedocs.io](https://mowgli.readthedocs.io/en/latest/models.html) (<https://mowgli.readthedocs.io/en/latest/models.html>) is missing a description of parameters such as `h_regularization`. The description is already present in Github. Also, documentation for `mowgli.score` api is missing as well. So, if it could be included in the documentation, it'll help a wider group of users.

Reply: We updated the documentation as suggested by the Reviewer.

2. Line 682: In the specificity formula, j , although obvious, is not defined.

Reply: We defined j in the sentence explaining the formula for more clarity.

REVIEWERS' COMMENTS

Reviewer #1 (Remarks to the Author):

I thank the authors for taking into account my additional comments. I feel that they addressed all remaining points.

Reviewer #2 (Remarks to the Author):

Thanks for the additional analysis, and I believe the manuscript is ready to move forward now.